# Renal antiporter ClC-5 regulates collagen I/IV through the *β*-catenin pathway and lysosomal degradation

Mònica Durán[1], Gema Ariceta[1,2,3], Maria E Semidey[4], Carla Castells-Esteve[1], Andrea Casal-Pardo[1], Baisong Lu[5], Anna Meseguer[1,6,*], Gerard Cantero-Recasens[1,*]

**Mutations in Cl⁻/H⁺ antiporter ClC-5 cause Dent's disease type 1 (DD1), a rare tubulopathy that progresses to renal fibrosis and kidney failure. Here, we have used DD1 human cellular models and renal tissue from DD1 mice to unravel the role of ClC-5 in renal fibrosis. Our results in cell systems have shown that ClC-5 deletion causes an increase in collagen I (Col I) and IV (Col IV) intracellular levels by promoting their transcription through the *β*-catenin pathway and impairing their lysosomal-mediated degradation. Increased production of Col I/IV in ClC-5–depleted cells ends up in higher release to the extracellular medium, which may lead to renal fibrosis. Furthermore, our data have revealed that 3-mo-old mice lacking ClC-5 (*Clcn5*⁺/⁻ and *Clcn5*⁻/⁻) present higher renal collagen deposition and fibrosis than WT mice. Altogether, we describe a new regulatory mechanism for collagens' production and release by ClC-5, which is altered in DD1 and provides a better understanding of disease progression to renal fibrosis.**

## Introduction

Renal fibrosis is a common pathological process occurring as a consequence of chronic kidney injury, independently of the underlying disease aetiology (1). It is characterized by pathological deposition and accumulation of ECM in the renal interstitium (2). Renal ECM is composed mainly of collagens, elastin, and other glycoproteins to form the basement membrane. In normal conditions, ECM is in a constant remodelling state, with an equilibrium between degradation of old or damaged molecules and replacement by new ones. In fibrosis, however, this is altered with increased production and reduced degradation, leading to excessive ECM deposition (3). Indeed, collagen turnover is a key event for ECM remodelling and impairment of this process leads

to fibrosis initiation and progression (4). Interestingly, up to 30% of newly synthesized collagen is sent for intracellular degradation (5, 6) as a mechanism to prevent defective collagen's secretion and therefore preserve the quality of ECM. In addition, collagen type I, II, III, IV, V, VI, VII, and XV levels are increased in renal fibrosis (1, 3).

ClC-5 (Cl⁻/H⁺ exchange transporter 5), encoded by the *CLCN5* gene, is mainly expressed in renal epithelial cells of the proximal tubule (PTC) (7). It controls endosomal acidification by cooperating with the V-ATPase, which affects protein recycling to the plasma membrane (e.g., endocytic receptors) (8). Accordingly, ClC-5 loss of function leads to alterations in endocytic activity (9) and delays protein progression to lysosomes, thereby affecting the functioning of PTCs (8). In fact, mutations in *CLCN5* cause Dent's disease type 1 (DD1), a rare X-linked renal proximal tubulopathy characterized by hypercalciuria and low molecular weight proteinuria (10, 11). In addition, some patients may show an incomplete or complete renal Fanconi syndrome (12). DD1 progresses to chronic kidney disease and finally to renal failure requiring renal replacement therapy between the thirties and fifties in up to 80% of patients (13). Importantly, several DD1 patients present tubulointerstitial fibrosis even with normal renal function (14). Besides, kidney biopsies from DD1 patients demonstrated progressive interstitial fibrosis associated with glomerular sclerosis overtime, in parallel to glomerular filtration decline (15). Interestingly, lack of ClC-5 has been shown to lead to epithelial cell dedifferentiation and decrease in endocytic receptors in both mouse and cell models (16, 17, 18, 19). However, although ClC-5 loss of function is the cause of DD1 and its role as Cl⁻/H⁺ exchanger is well studied, the link between ClC-5 and renal fibrosis remains unknown.

Here, we show that ClC-5 modulates intracellular and extracellular collagen levels by controlling their synthesis through the *β*-catenin pathway and their lysosomal degradation. Impairment of ClC-5 function alters the equilibrium between these processes, increasing collagen levels and thus contributing to renal fibrosis.

[1]Renal Physiopathology Group, Vall d'Hebron Research Institute (VHIR), Barcelona, Spain   [2]Pediatric Nephrology Department, Vall d'Hebron University Hospital, Barcelona, Spain   [3]Pediatrics Department, School of Medicine, Autonomous University of Barcelona (UAB), Bellaterra, Spain   [4]Department of Pathology, Vall d'Hebron University Hospital, Barcelona, Spain   [5]Wake Forest Institute for Regenerative Medicine, Winston-Salem, NC, USA   [6]Biochemistry and Molecular Biology Department, School of Medicine, Autonomous University of Barcelona (UAB), Bellaterra, Spain

Correspondence: ana.meseguer@vhir.org; gerard.cantero@vhir.org
*Anna Meseguer and Gerard Cantero-Recasens are joint last authors

  

# Results

## ClC-5 controls the production and release of collagen types I and IV in renal cells

Uncontrolled collagens' release and accumulation at tubulointerstitial space is characteristic of renal fibrosis, a process occurring in most Dent's disease type 1 (DD1) patients (14). Nonetheless, whether and how ClC-5, the genetic cause of DD1, may regulate collagens' levels has not been addressed.

Here, we have used renal proximal tubule epithelial cell lines (RPTEC/TERT1) depleted of endogenous *CLCN5* (*CLCN5* KD) or overexpressing ClC-5 WT rescue form (rWT) previously generated by our laboratory (18) to study the role of ClC-5 in collagen production and secretion. First, mRNA and protein were extracted from each cell line after 10 d of differentiation (as described in reference 18) to validate them. Analysis of *CLCN5*/ClC-5 levels confirmed a >90% reduction in *CLCN5* KD cells and complete expression recovery in rWT cells compared with control cells (Fig S1A and B). Next, we studied whether intracellular and extracellular protein levels of collagen types I and IV (Col I and Col IV), which are key elements of renal fibrotic tissues, were affected by ClC-5 depletion. Briefly, total cell lysate and secreted medium from control, *CLCN5* KD, and ClC-5 rWT differentiated cells were collected and Col I/IV levels analysed by Western blot (WB). Our results revealed that ClC-5 deletion causes a strong increase in intracellular Col I and Col IV levels compared with control cells (fourfold and 10-fold increase, respectively). Similarly, Col I/IV extracellular levels were also elevated in *CLCN5* KD cells compared with control cells (sixfold and fourfold increase, respectively). Importantly, the expression of the ClC-5 rWT form rescued the phenotype of *CLCN5* KD (Fig 1A–D). Next, we studied the effect of different pathogenic ClC-5 mutants (V523del, E527D, and I524K) on the *CLCN5* KD collagen IV phenotype. These mutants were correctly expressed at the mRNA level in respective cell lines (Fig S1A), although protein levels were more variable because of mutants being delivered for degradation as previously described (Fig S1C) (18). Interestingly, our data showed that the expression of V523del, E527D, and I524K mutants restored both Col IV intracellular levels to those of the rWT condition, but did not rescue the *CLCN5* KD phenotype on Col IV extracellular levels (or partially, in the case of V523del and I524K) (Fig S2A). This reflects the loss-of-function differences between ClC-5 mutants and may contribute to DD1 phenotypic variability (20). To sum up, these data show that only the WT form of ClC-5, but not the pathogenic mutants, can fully rescue the *CLCN5* KD phenotype.

Then, to confirm the increase in collagen levels caused by ClC-5 depletion, differentiated control, *CLCN5* KD, and ClC-5 rWT cells were permeabilized, processed for immunofluorescence with anti-Col I/IV antibodies, and imaged by confocal microscopy. Our data showed that *CLCN5* KD cells present a massive increase in intracellular Col I and Col IV signals compared with control or rWT cells (Fig 1E). Then, to visualize extracellular Col I and IV, mildly washed and not permeabilized cells were fixed and imaged by confocal microscopy. In accordance with WB results, we found that *CLCN5* KD cells present a higher than 2.5-fold increase in Col

I/IV extracellular fluorescence compared with control and rWT cells (Fig 1F, quantification in Fig 1G), which correlated with detection of more Col I/IV extracellular fibres (Col I: 11.7 versus 2.7 and 3.3 fibres; Col IV: 13.5 versus 5.2 and 2.0 fibres, KD versus control and rWT cells, respectively) (Fig 1H). Next, to further assess whether this phenotype of *CLCN5* KD cells is specific for Col I/IV intracellular and extracellular levels, we studied two individual cargo proteins, fibronectin 1 (FN1, component of the ECM) and cathepsin D (CTSD, typical lysosomal enzyme that can be released to the extracellular medium) (21, 22). Notably, our analyses showed no major effect of *CLCN5* KD on FN1 intracellular or extracellular levels. CTSD intracellular levels, however, were strongly reduced in cells depleted of ClC-5 (75% decrease), a phenotype mildly rescued by ClC-5 rWT expression. Similarly, CTSD release to the extracellular medium was totally abolished in *CLCN5* KD cells compared with control cells (93% reduction) (Fig S2B). This fits previous microarray data showing no differences in *FN1* mRNA levels between control and *CLCN5* KD cells, but a strong downregulation of *CTSD* in cells lacking a functional ClC-5 (18). In conclusion, our results demonstrate that *CLCN5* depletion specifically affects collagen I/IV levels and does not have a general effect on secretory cargoes.

## Col I/IV accumulate intracellularly at the ER and lysosomes of *CLCN5* KD cells

Deletion of *CLCN5* causes a massive increase in Col I and IV intracellular levels. However, how is ClC-5 affecting collagens' levels? Is ClC-5 promoting collagen production or preventing their degradation? To elucidate the mechanism, we started by studying where do collagens accumulate in *CLCN5* KD cells. Differentiated control, *CLCN5* KD, and ClC-5 rWT cells were processed for immunofluorescence, costained with anti-Col I/IV and the endoplasmic reticulum (ER) marker KDEL, Golgi apparatus marker GM130, or the lysosomal marker LAMP1, and imaged by confocal microscopy. Our results showed major colocalization of Col I and IV with KDEL in *CLCN5* KD cells compared with control or rWT cells (17.5- and 10.5-fold increase, respectively, *P* < 0.001) (Figs 2A and B and S3A and B). In addition, we also detected higher colocalization of Col I and IV with LAMP1 in *CLCN5* KD cells than control or ClC-5 rWT cells (3- and 3.5-fold increase, *P* < 0.05) (Figs 2C and D and S3C and D). Importantly, colocalization of both Col I with GM130 was similar in all conditions, although it was mildly higher in *CLCN5* KD cells for Col IV (Fig S4A–D). Altogether, our results confirmed that collagens mostly accumulate in the ER of *CLCN5* KD, and at a lower degree at the lysosomes, suggesting an effect on collagens' degradation. To further assess this possibility, we also analysed collagens' localization after treatment with bafilomycin A1 (inhibitor of v-ATPase that impairs lysosomal activity). Our results showed no major effects on Col I/IV colocalization with KDEL, but increased the colocalization with LAMP1 in control and ClC-5 rWT cells, and to a lesser extent in *CLCN5* KD cells (Figs 2A and B and S3A and B). Notably, Col I colocalization with GM130 was strongly reduced after bafilomycin treatment, which fits with higher accumulation in lysosomes and impaired intracellular trafficking (Fig S4A).

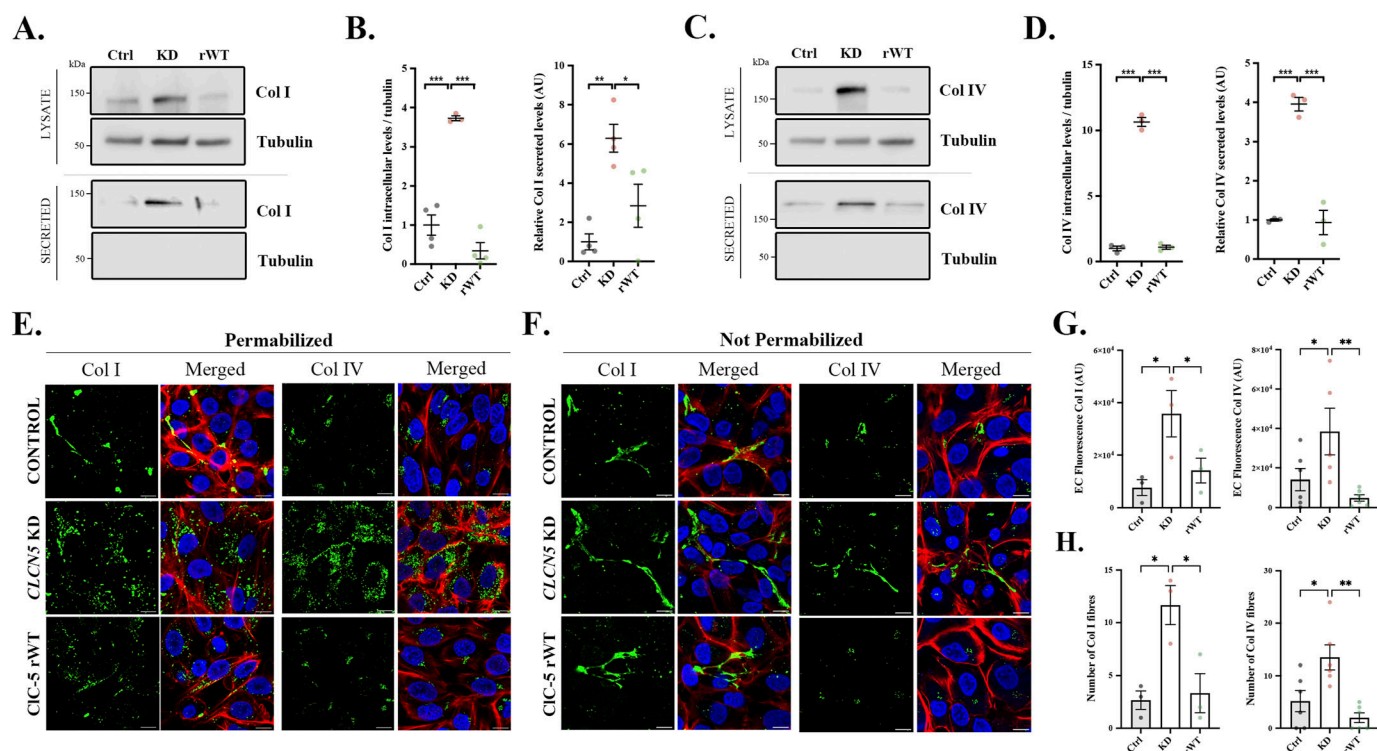

**Figure 1. Collagen type I and type IV intracellular and extracellular levels are increased in *CLCN5* KD cells.**
**(A, C)** Collagen type I (Col I) (A) or collagen type IV (Col IV) (C) protein levels in the lysates or secreted media of control (Ctrl), *CLCN5* KD (KD), and ClC-5 rWT (rWT) cells. Tubulin was used as a loading control in the lysates, and as a control of lysate contamination in secreted media. **(B, D)** Quantification of Col I (B) or Col IV (D) protein intracellular levels (left graph) or secreted levels (right graph) of control (Ctrl, black dots), *CLCN5* KD (KD, pink dots), and ClC-5 rWT (rWT, green dots) cells (N ≥ 3). All values were normalized to tubulin levels. **(E)** Representative immunofluorescence z-stack single planes of permabilized control, *CLCN5* KD, and ClC-5 rWT cells stained with anti-Col I or anti-Col IV antibody (green), phalloidin (red), and Hoechst 33342 (blue). Scale bars: 5 *μ*m. **(F)** Representative immunofluorescence z-stack single planes of not permeabilized control, *CLCN5* KD, and ClC-5 rWT cells stained with anti-Col I or anti-Col IV antibody (green), phalloidin (red), and Hoechst 33342 (blue). Scale bars: 5 *μ*m. **(G)** Quantification of Col I and IV extracellular fluorescence from not permeabilized control, *CLCN5* KD, and ClC-5 rWT cells (N ≥ 3). **(H)** Quantification of Col I and IV fibres from not permeabilized control, *CLCN5* KD, and ClC-5 rWT cells (N ≥ 3). Abbreviations: Col I, collagen type I; Col IV, collagen type IV; Ctrl, control cells; KD, *CLCN5* KD cells; rWT, ClC-5 rWT cells; EC, extracellular. Statistical significance was determined using one-way ANOVA followed by Tukey's post hoc test. *P < 0.05, **P < 0.01, ***P < 0.001. Source data are available for this figure.

## ClC-5 deletion impairs Col I and Col IV degradation

It is known that 30% of newly synthesized collagen is normally degraded (23). Therefore, considering that ClC-5 is a key component of the endolysosomal system, and the accumulation of collagens in the lysosomes of *CLCN5* KD cells or bafilomycin A1–treated cells, we postulated that the increase in collagens' intracellular levels may be caused by impairment of their degradation. Thus, we proceeded to study Col I/IV half-life in control, *CLCN5* KD, and ClC-5 rWT cells by a cycloheximide assay. As shown in Fig 3, Col I and IV half-life was longer in *CLCN5* KD cells compared with control cells (11.2 versus 2.2 h and 13.1 versus 1.75 h, respectively, *P* < 0.05). The overexpression of ClC-5 rWT completely rescued this phenotype (Col I: 3.3 h and Col IV: 2.5 h) (Fig 3A and B). Altogether, our data indicate that collagens' degradation is affected by ClC-5 deletion.

To understand whether ClC-5 depletion affects lysosomal or proteasomal degradation, we then studied how collagens are degraded in RPTEC/TERT1 using inhibitors of the proteasome (MG132) or lysosomes (leupeptin and bafilomycin A1). Our results showed that only inhibitors of lysosomes caused Col I and IV intracellular accumulation compared with vehicle in control cells

(leupeptin or bafilomycin A1: Col I, 4.1- and 2.3-fold increase; Col IV: 4.8- and 3.0-fold increase, respectively, *P* < 0.05) (Fig 3C). Chemical inhibition of lysosomes by leupeptin, which caused intracellular accumulation of Col I/IV, also produced an increase of extracellular collagen levels in control cells. Interestingly, treatment with bafilomycin, which is known to inhibit exocytosis (24), did not cause a major release of collagens (Fig S5A). Next, to confirm that lysosomal degradation of collagens was impaired because of ClC-5 deletion, we treated *CLCN5* KD cells with lysosomal inhibitors and analysed Col IV levels. We found that neither leupeptin (inhibitor of lysosomal enzymes) nor bafilomycin A1 (inhibitor of v-ATPase) further increased Col IV intracellular or extracellular levels in *CLCN5* KD cells (Fig 3D). These data suggest that ClC-5 impairs collagen degradation by altering lysosomal activity.

## *CLCN5* depletion affects lysosomal acidification

Our data demonstrate that Col I/IV are mainly degraded through lysosomes in RPTEC/TERT1 cells, a process impaired in cells lacking ClC-5. Other authors have shown that ClC-5 has a role in endosomal acidification (8); however, is ClC-5 also affecting lysosomal pH in

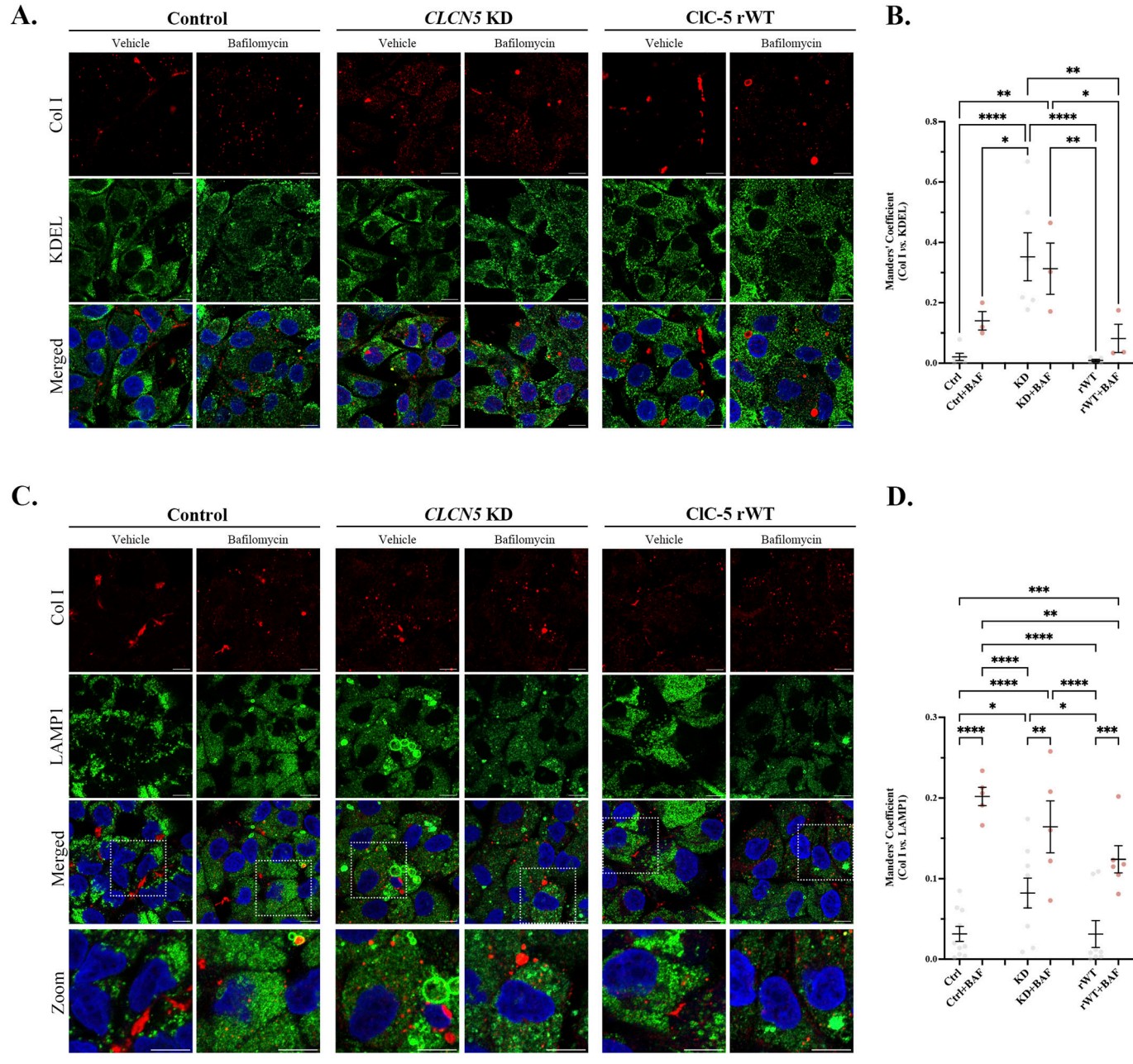

**Figure 2.  Col I accumulates in the endoplasmic reticulum and lysosomes of *CLCN5* KD cells.**
**(A)** Representative immunofluorescence z-stack single-plane images of control, *CLCN5* KD, and ClC-5 rWT cells treated with vehicle or bafilomycin A1 and stained with anti-Col I antibody (red), KDEL (green), and Hoechst 33342 (blue). Scale bars: 5 μm. **(B)** Colocalization between Col I and KDEL for different cell lines (control, *CLCN5* KD, and ClC-5 rWT) was calculated from immunofluorescence images by Manders' coefficient using FIJI (N ≥ 3). Average values ± SEM are plotted as a scatter plot with a bar graph. The y-axis represents Manders' coefficient of the fraction of Col I overlapping with KDEL. **(C)** Representative immunofluorescence z-stack single-plane images of control, *CLCN5* KD, and ClC-5 rWT cells treated with vehicle or bafilomycin A1 and stained with anti-Col I antibody (red), LAMP1 (green), and Hoechst 33342 (blue). Zoomed images are provided to show Col I and LAMP1 close localization. White dashed line squares represent the zoomed area. Scale bars: 5 μm. **(D)** Colocalization between Col I and LAMP1 for different cell lines (control, *CLCN5* KD, and ClC-5 rWT) was calculated from immunofluorescence images by Manders' coefficient using FIJI (N ≥ 3). Average values ± SEM are plotted as a scatter plot with a bar graph. The y-axis represents Manders' coefficient of the fraction of Col I overlapping with LAMP1. Abbreviations: Col I, collagen type I; Ctrl, control cells; KD, *CLCN5* KD cells; rWT, ClC-5 rWT cells; BAF, bafilomycin A1. Statistical significance was determined using one-way ANOVA followed by Tukey's post hoc test. *$P < 0.05$, ***$P < 0.001$.

RPTEC/TERT1 cells? To test this possibility, we measured the luminal lysosomal pH using the LysoSensor Yellow/Blue DND-160 dye. Colocalization of LysoSensor with LAMP1 showed that after 5 min, this dye is mostly found in the lysosomes (Fig S5B). Thus, all subsequent experiments were performed after 5-min incubation to ensure LysoSensor is at lysosomes. Our data showed that in basal conditions, lysosomes' luminal pH of *CLCN5* KD cells is more basic than that of control cells (20% increase, $P < 0.01$), and is partially

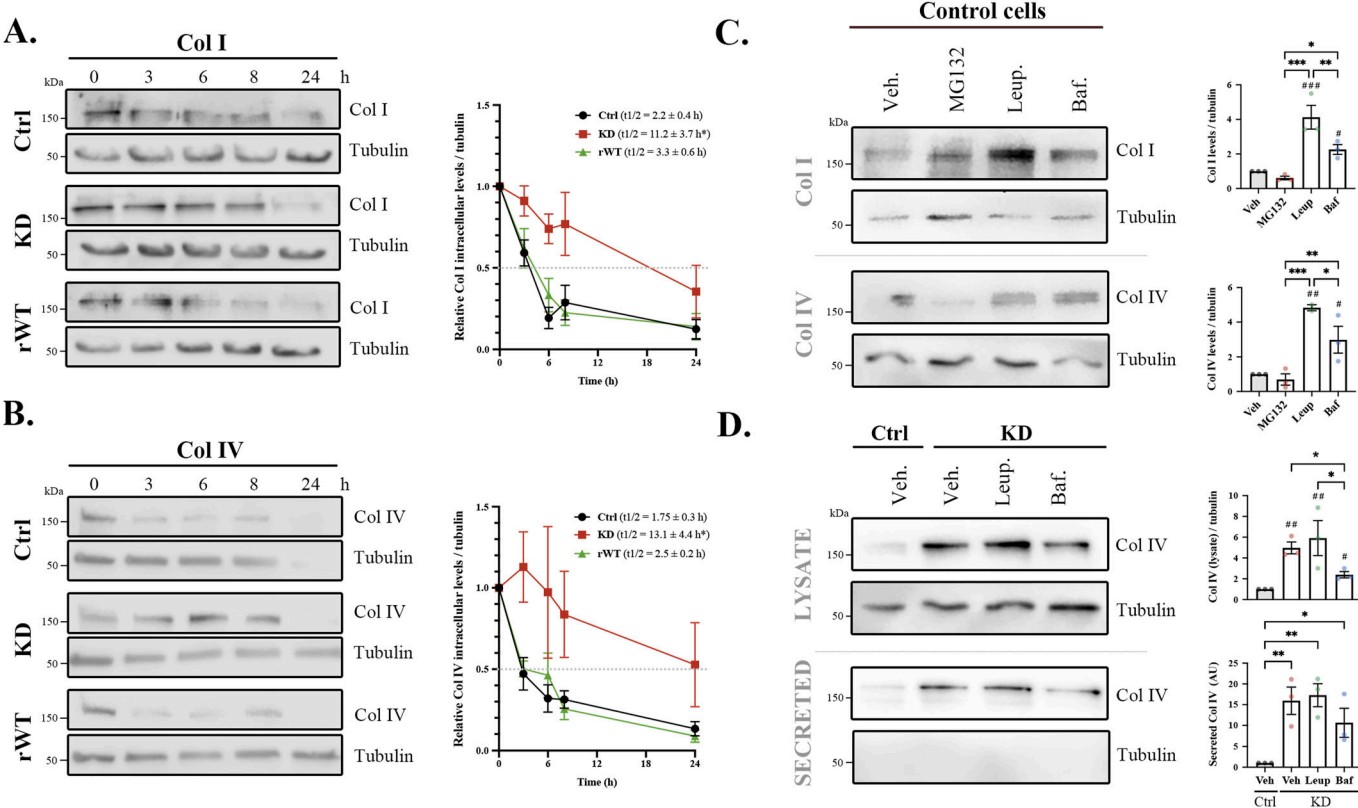

**Figure 3. Collagen lysosomal degradation is impaired in cells lacking *CLCN5*.**
**(A, B)** Representative blots of cycloheximide chase assay for Col I (A) and Col IV (B) protein levels over time (0, 3, 6, 8, and 24 h) of control (Ctrl), *CLCN5* KD (KD), and ClC-5 rWT (rWT) cells. Quantification of respective protein levels for each time point normalized to tubulin and relative to time = 0 h is shown in the graph. The half-life of Col I or Col IV is shown for each cell line (N > 3). Statistical significance was determined using one-way ANOVA followed by Tukey's post hoc test. *P < 0.05 (*CLCN5* KD compared with control and ClC-5 rWT cells). **(C)** Western blots of Col I (upper lanes) or Col IV (lower lanes) from lysates of control cells treated with vehicle (Veh.), MG132, leupeptin (Leup.), or bafilomycin (Baf.). Tubulin was used as a loading control. Quantification of Col I and Col IV is shown in the graphs for Col I and IV (N = 3). **(D)** Col IV intracellular (lysate) or extracellular (secreted) levels from control cells (Ctrl) or *CLCN5* KD cells (KD) treated with vehicle (Veh.), leupeptin (Leup.), or bafilomycin (Baf.). Tubulin was used as a loading control. Quantification of intracellular and extracellular levels of Col IV is shown in the graphs (N = 3). Abbreviations: Col I, collagen type I; Col IV, collagen type IV; Ctrl, control cells; KD, *CLCN5* KD cells; rWT, ClC-5 rWT cells; Veh, vehicle; Leup, leupeptin; Baf, bafilomycin. Statistical significance was determined using one-way ANOVA followed by Tukey's post hoc test. *P < 0.05, ***P < 0.001, #P < 0.05, ##P < 0.01 compared with control vehicle.
Source data are available for this figure.

recovered by the expression of ClC-5 rWT (Fig 4A). To reinforce these results, we also tested lysosomal enzymatic activity using Magic Red Cathepsin B Kit (#ICT938; Bio-Rad). Our data revealed a 30% decrease in Magic Red fluorescence in *CLCN5* KD compared with control cells, which was rescued in ClC-5 rWT cells. Treatment with bafilomycin A1 (as a positive control) reduced lysosomal activity to a similar level for all conditions (35% decrease in Magic Red fluorescence) (Fig 4B). Next, we studied whether the localization, size, or number of lysosomes were altered in cells lacking ClC-5, which could be affected because of impaired lysosomal degradation of collagens. Immunofluorescence studies using anti-LAMP1 antibody and anti-KDEL antibody revealed a higher colocalization between both markers in *CLCN5* KD cells than in control or rWT cells (twofold increase, P < 0.05) (Fig 4C and D). Although the total number of lysosomes was not statistically different between cell lines, further analysis of their distribution revealed a higher concentration of lysosomes in the perinuclear area of *CLCN5* KD cells compared with control and ClC-5 rWT cells (55% versus 44% and 42%, respectively, P < 0.01) (Fig 4E). Notably, our results showed an

increased percentage of big lysosomes in *CLCN5* KD cells compared with control or rWT cells (29.2% ± 1.7%, 18.6% ± 2.3%, and 19.4% ± 0.4%, respectively) (Fig 4F). Volume measurement also showed that lysosomes from *CLCN5* KD cells were bigger than control or rWT cells (median volume of KD: 0.18 $\mu m^3$, control: 0.11 $\mu m^3$, rWT: 0.10 $\mu m^3$, P < 0.01). These results fit with impaired degradation of collagens, which accumulate intracellularly resulting in bigger lysosomes that remain closer to the ER.

Altogether, these data confirm that ClC-5 depletion impairs lysosomal acidification, leading to defects in collagen degradation and causing an increase in Col I/IV intracellular levels.

## ClC-5 deletion promotes Col I/IV transcription through the β-catenin pathway

Impairment of lysosomal degradation can account for the collagen accumulation in lysosomes of *CLCN5* KD cells, but cannot explain the enhanced levels at the ER. Because proteasomal degradation was not altered, we postulated that Col I and Col IV synthesis could

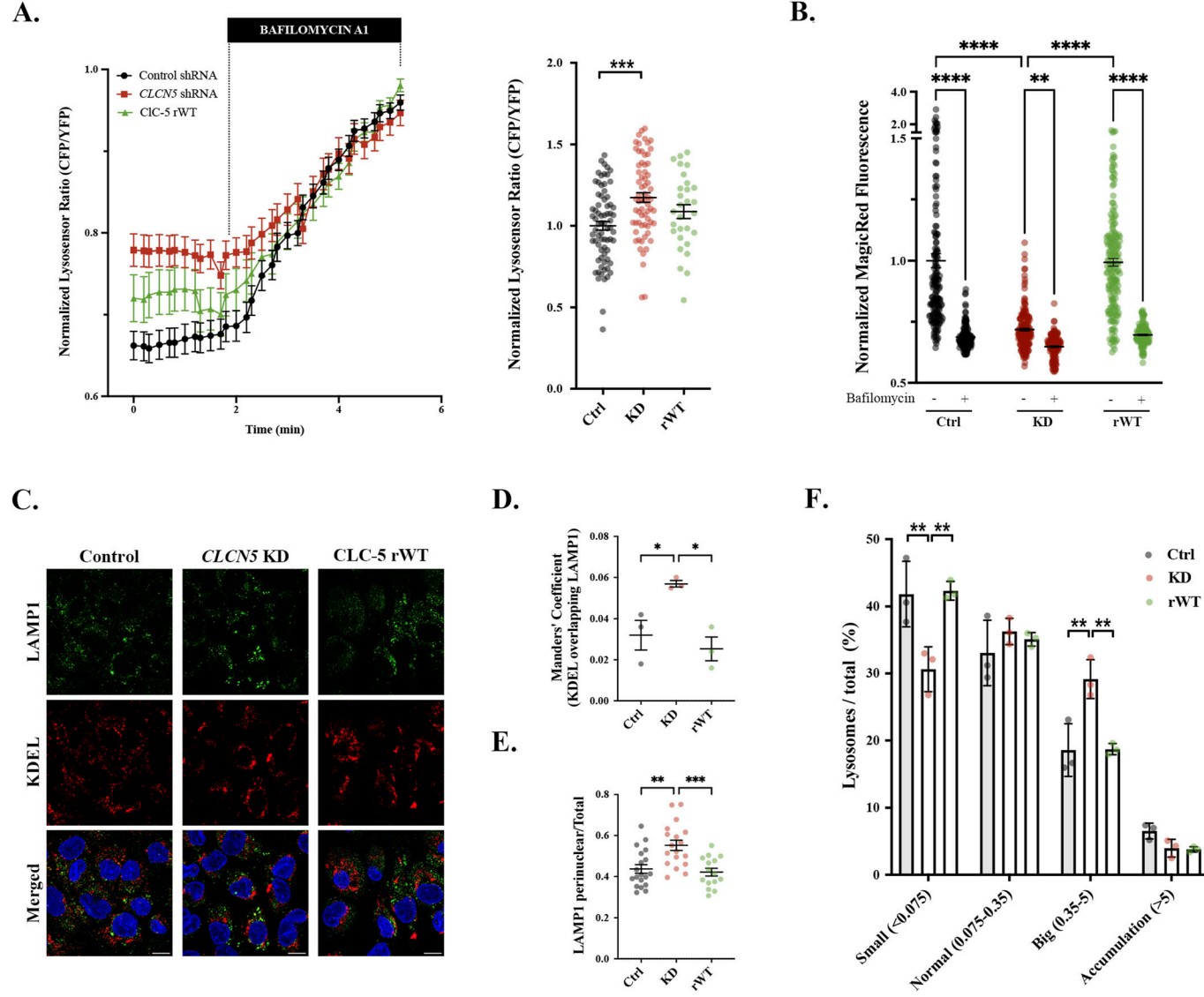

**Figure 4. *CLCN5* KD cells present less acidic and larger lysosomes than control cells.**
**(A)** Time course of the CFP/YFP LysoSensor Yellow/Blue DND-160 dye ratio in control (control shRNA, black), *CLCN5* KD (*CLCN5* shRNA, red), and ClC-5 rWT cells. After 2 min of basal activity, bafilomycin was added as a normalization control. Quantification of the normalized basal CFP/YFP LysoSensor Yellow/Blue DND-160 dye ratio is shown in the left graph. Each dot represents one cell (N ≥ 3). **(B)** Quantification of the lysosomal activity assay using Magic Red Cathepsin B Kit for control, *CLCN5* KD, or ClC-5 rWT cells treated with or without bafilomycin A1 (N ≥ 5). **(C)** Representative immunofluorescence z-stack single-plane images of control, *CLCN5* KD, and ClC-5 rWT cells stained with anti-LAMP1 (green), anti-KDEL (red), and Hoechst 33342 (blue). Scale bars: 5 *μm*. **(D)** Colocalization between LAMP1 and KDEL for different cell lines (control, *CLCN5* KD, and ClC-5 rWT) was calculated from immunofluorescence images by Manders' coefficient using FIJI (N ≥ 3). Average values ± SEM are plotted as a scatter plot with a bar graph. The y-axis represents Manders' coefficient of the fraction of KDEL overlapping with LAMP1. **(E)** Quantification of LAMP1 perinuclear distribution in control, *CLCN5* KD, or ClC-5 rWT cells. The perinuclear area was defined as the region extending 0.5x the nuclear radii (R and r for short and large radius, respectively) (N ≥ 3). **(F)** Percentage of the populations of lysosomes of control (Ctrl), *CLCN5* KD (KD), and ClC-5 rWT (rWT) cells classified by volume: small (<0.075 *μm³*), normal (between 0.075 and 0.35 *μm³*), big (between 0.35 and 5 *μm³*), and accumulations (>5 *μm³*). The volume of lysosomes was calculated from individual immunofluorescence images of LAMP1 using 3D analysis Fiji software (N ≥ 3). Abbreviations: Col I, collagen type I; Col IV, collagen type IV; Ctrl, control cells; KD, *CLCN5* KD cells; rWT, ClC-5 rWT cells. Statistical significance was determined using one-way ANOVA followed by Tukey's post hoc test. *P < 0.05, **P < 0.01, ***P < 0.001.

be enhanced in *CLCN5*-depleted cells. To test this hypothesis, first we tested Col I and IV mRNA levels in our cell lines. Our data revealed that Col I/IV mRNA levels, both increased in renal fibrosis, were higher in *CLCN5* KD cells than in control or rWT cells (4.9- and 4.0-fold increase, respectively, *P* < 0.01). Indeed, a similar increment was detected after TGF-*β* stimulation on control cells, a classic stimulator of collagen transcription

through the *β*-catenin pathway, suggesting its involvement in the control of collagen expression by ClC-5. Notably, TGF-*β* treatment further potentiated *CLCN5* KD effect on Col I/IV transcription, suggesting a summatory effect possibly through other TGF-*β*–activated pathways (Fig 5A). It has been extensively described that *β*-catenin release from the plasma membrane (PM) and translocation to the nucleus facilitate epithelial–mesenchymal

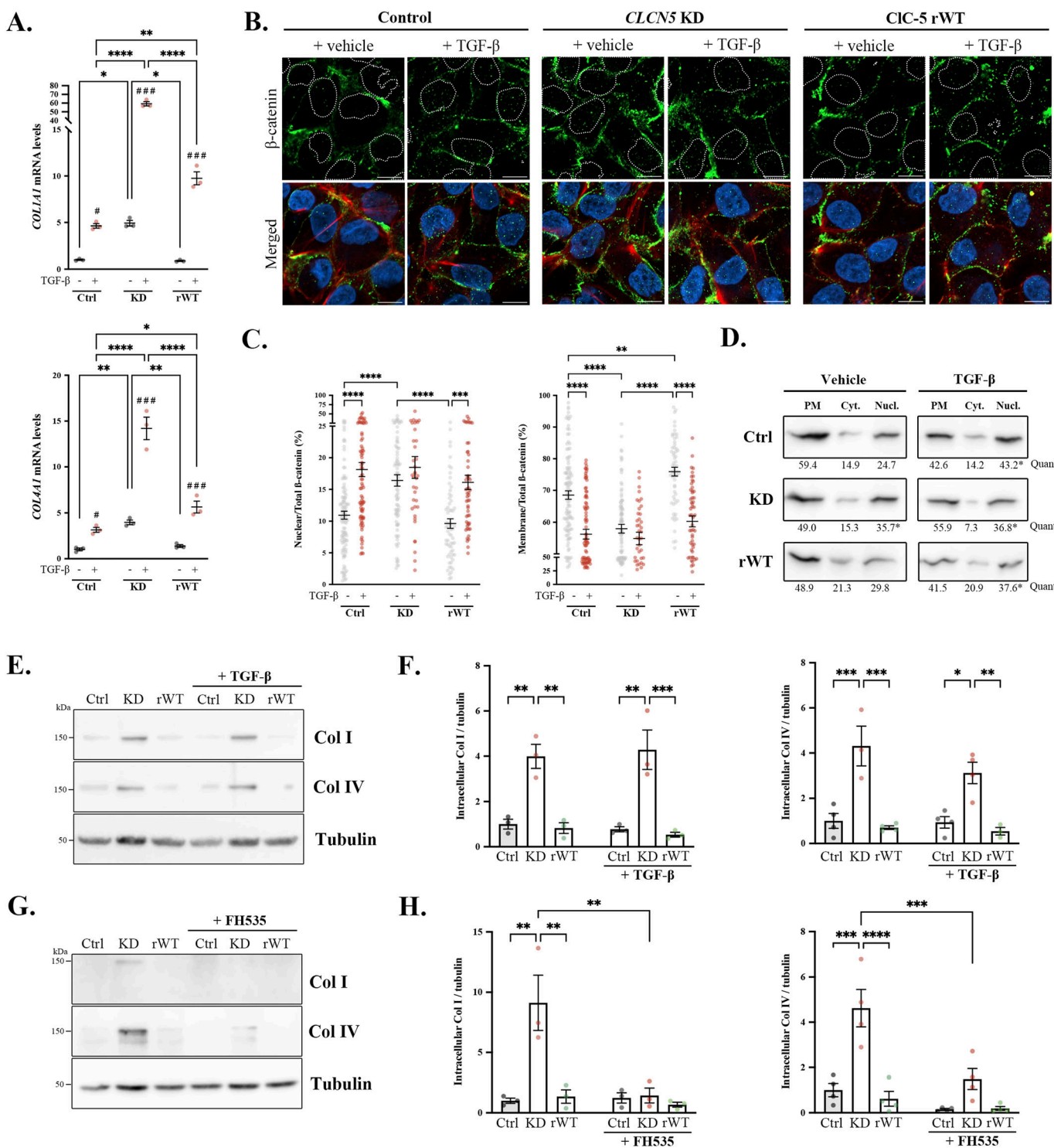

**Figure 5. *CLCN5* KD cells present higher β-catenin nuclear translocation.**
**(A)** *COL1A1* (Col I) and *COL4A1* (Col IV) mRNA levels (normalized to *HPRT1* levels) from control (black dots), *CLCN5* KD (pink dots), and ClC-5 rWT (green dots) cells treated with or without TGF-β. **(B)** Immunofluorescence single planes of control, *CLCN5* KD, and ClC-5 rWT cells (treated with vehicle or TGF-β) stained with anti-β-catenin (green), phalloidin (red), and Hoechst 33342 (blue). The nucleus perimeter is demarcated with dotted white lines to visualize β-catenin nuclear translocation. Scale bars: 5 μm. **(C)** β-Catenin levels at the nucleus (left graph) or at the membrane (right graph) normalized to total levels were quantified from immunofluorescence images of vehicle (grey dots)- or TGF-β (red dots)–treated control, *CLCN5* KD, and ClC-5 rWT cells (N ≥ 3). **(D)** Representative β-catenin blots from the plasma membrane, cytosol, and nuclear fractions of control, *CLCN5* KD, and ClC-5 rWT cells treated with or without TGF-β. Quantification of the percentage of each fraction of the respective total from a minimum of three independent experiments is provided (N ≥ 3). *P < 0.05, nuclear percentage of β-catenin compared with control vehicle situation. **(E)** Collagen type I (Col I) and collagen type IV (Col IV) protein levels in the lysates of control (Ctrl), *CLCN5* KD (KD), and ClC-5 rWT (rWT) cells treated with vehicle or 1.5 ng/ml TGF-β. Tubulin was used as a

transition, promoting collagen transcription (25, 26) and leading to renal fibrosis (27, 28, 29). Hence, is β-catenin localization altered in *CLCN5* KD cells, which may explain the increase in collagen mRNA levels?

Immunofluorescence analyses revealed that, as expected, β-catenin is mostly localized at PM in control cells and is translocated to the nucleus after stimulation by the classic pathway activator TGF-β (30) (1.7-fold increase, *P* < 0.01). Interestingly, β-catenin was also more often found at the nucleus in *CLCN5* KD cells compared with control cells, which was prevented by ClC-5 rWT expression (KD: 1.5-fold increase, *P* < 0.01; rWT: 0.9-fold increase, n.s., compared with control cells). Treatment with TGF-β triggered higher nuclear translocation of β-catenin in ClC-5 rWT cells (1.7-fold increase, *P* < 0.01), but did not cause a major effect in ClC-5–depleted cells (Fig 5B, quantification in Fig 5C). To validate these data on β-catenin nuclear translocation, we performed a fractionation assay of control, *CLCN5* KD, and ClC-5 rWT cell lysates. Our findings showed a clear increase in β-catenin in the nuclear fraction of *CLCN5* KD cells (control: 24.7%, KD: 35.7%, rWT: 29.8%, *P* < 0.05) or TGF-β–treated cells compared with control and ClC-5 rWT cells (control-treated: 43.2%, KD-treated: 36.8%, rWT-treated: 37.6%, *P* < 0.05) (Fig 5D). Next, we studied whether this translocation promoted the expression of known β-catenin target genes (31). Analysis of the public data from DD1 cellular model microarrays (18) showed that several of β-catenin target genes were also modulated by *CLCN5* deletion (Table S1). Four representative genes (*BMP4*, *SOX2*, *SOX9*, and *EFNB1*) were selected to validate these data. Accordingly, *SOX2* and *BMP4* were up-regulated (eightfold and 2.5-fold increase, respectively), whereas *SOX9* and *EFNB1* were down-regulated (80% and 75% reduction, respectively) in *CLCN5* KD cells. ClC-5 rWT expression partially recovered *BMP4* and *EFNB1* levels, suggesting that other pathways rather than β-catenin may be regulating the expression of the other genes (Fig S6A).

To further assess the contribution of β-catenin to ClC-5 regulation of collagen levels, we used the pathway stimulator TGF-β and the inhibitor of β-catenin translocation FH535 (30, 32). Surprisingly, TGF-β treatment, which clearly promotes collagen transcription, did not cause an increase of Col I/IV intracellular levels in any of the cell lines (Fig 5E, quantification in Fig 5F). Extracellular Col I/IV levels, however, were clearly incremented in all TGF-β–treated cells compared with respective vehicle-treated cells (Fig S6B). A reasonable explanation is that in normal conditions, stimulation of collagen production does not cause intracellular accumulations because newly synthesized collagens will be secreted or normally degraded. In *CLCN5* KD cells, TGF-β does not significantly increase intracellular levels, indicating that they act through the same pathway. Then, we tested the effect of FH535, which prevented β-catenin translocation (Fig S6C), on Col I/IV levels. We found that FH535 treatment strongly reduced Col I/IV intracellular and extracellular levels in *CLCN5* KD cells (Figs 5G and H and S6D). In summary, inhibition of β-catenin translocation reduces collagen production and, thus, causes a strong decrease of collagen levels in *CLCN5* KD cells.

Altogether, our data indicate that lack of *CLCN5* promotes collagen synthesis through the β-catenin pathway and, in combination with lysosomal degradation impairment, causes a huge intracellular collagen accumulation. Besides, our results suggest that this increased synthesis of new collagens ends in higher release to the extracellular medium.

## ClC-5 regulates mucin-1 (MUC1) levels in renal tubule epithelial cells

Cells lacking a functional ClC-5 present several characteristics of epithelial dedifferentiation (18), which we postulate promotes the increase in collagens' production and could explain renal fibrosis progression in DD1. In fact, E-cadherin or mucin-1 (MUC1), which are relevant cell polarization and differentiation markers (33), can form complexes with β-catenin, and their loss can lead to β-catenin translocation, thereby promoting collagen transcription (34). Confirming previous results (18), *CLCN5*-depleted cells showed a strong reduction in E-cadherin protein levels compared with control and rWT cells (>90% decrease, *P* < 0.05) (Fig 6A). We also analysed the levels of mucin-1 (MUC1), which was one of the most affected genes by ClC-5 loss of function as shown by microarray data (18). Accordingly, *CLCN5* deletion caused a decrease in MUC1 protein levels compared with control cells, which was partially recovered by the expression of ClC-5 rWT (Fig 6B). Confocal immunofluorescence analyses confirmed the reduction of MUC1 levels by *CLCN5* deletion, which was almost completely restored by the expression of ClC-5 rWT (Fig 6C). We also analysed the levels of MUC1 in ClC-5 mutant cell lines. Our results showed that only the rWT form of ClC-5 can restore normal levels of MUC1, whereas all ClC-5 loss-of-function forms maintain low levels of MUC1, as shown by Western blot and immunofluorescence (Fig S7A and B).

Next, to assess the relationship between MUC1 and β-catenin, we studied the effect of TGF-β or FH535 treatment on MUC1 levels. Our results showed that, on the one hand, inhibition of β-catenin translocation slightly reduced MUC1 levels in control cells (39% decrease, *P* < 0.05) (Fig 6D). On the other hand, stimulation with TGF-β caused an increase of MUC1 levels in control cells (2.7-fold increase, *P* < 0.0001) (Fig 6E), which supports previous studies regarding a feedback regulatory loop between MUC1 and β-catenin pathways (34). However, neither inhibition nor stimulation of the β-catenin pathway rescued *CLCN5* KD reduction of MUC1 levels, thus suggesting that this effect is independent of β-catenin translocation. Indeed, MUC1

---

loading control in the lysates. **(F)** Quantification of Col I (left) and Col IV (right) protein intracellular levels of control (Ctrl, black dots), *CLCN5* KD (KD, pink dots), and ClC-5 rWT (rWT, green dots) cells treated with vehicle or 1.5 ng/ml TGF-β (N ≥ 3). **(G)** Collagen type I (Col I) and collagen type IV (Col IV) protein levels in the lysates of control (Ctrl), *CLCN5* KD (KD), and ClC-5 rWT (rWT) cells treated with vehicle or 10 μM FH535. Tubulin was used as a loading control in the lysates. **(H)** Quantification of Col I (left) and Col IV (right) protein intracellular levels of control (Ctrl, black dots), *CLCN5* KD (KD, pink dots), and ClC-5 rWT (rWT, green dots) cells treated with vehicle or 10 μM FH535 (N ≥ 3). Abbreviations: Ctrl, control cells; KD, *CLCN5* KD cells; rWT, ClC-5 rWT cells; Quant, quantification. Statistical significance was determined using one-way ANOVA followed by Tukey's post hoc test. *P* < 0.05, **P* < 0.01, ***P* < 0.001, ****P* < 0.0001.
Source data are available for this figure.

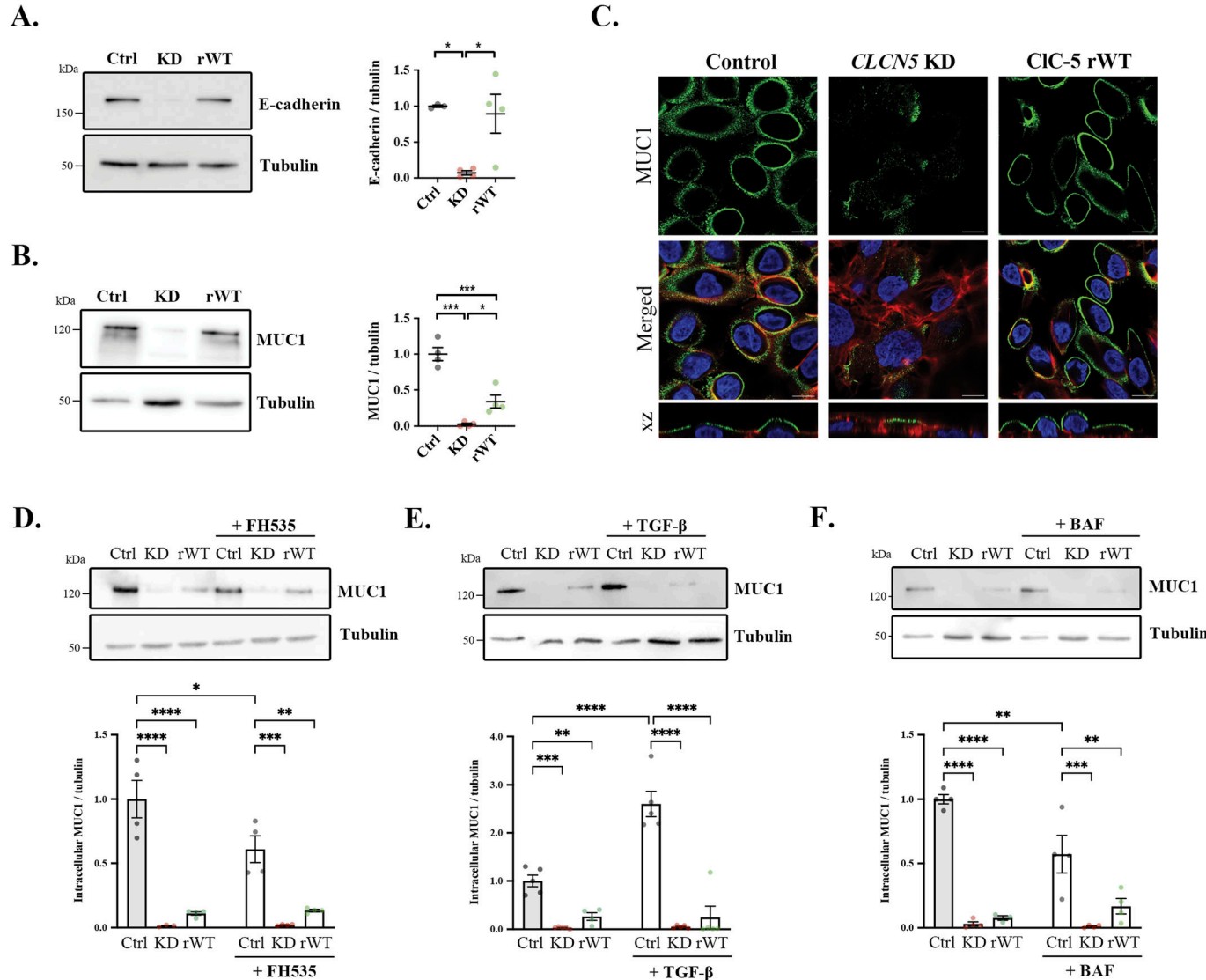

**Figure 6. Lack of *CLCN5* causes a decrease of MUC1 levels.**
**(A)** E-cadherin protein levels were analysed by Western blot in cell lysates from control, *CLCN5* KD, and ClC-5 rWT cells. Tubulin was used as a loading control. The right graph shows the quantification of E-cadherin levels (N > 3). **(B)** MUC1 protein levels were analysed by Western blot in cell lysates from control, *CLCN5* KD, and ClC-5 rWT cells. Tubulin was used as a loading control. The right graph shows the quantification of MUC1 levels (N > 3). **(C)** Immunofluorescence z-stack single planes of control, *CLCN5* KD, and ClC-5 rWT cells stained with anti-MUC1 antibody (green), phalloidin (red), and Hoechst 33342 (blue). Scale bars: 5 μm. **(D)** MUC1 protein levels in the lysates of control (Ctrl), *CLCN5* KD (KD), and ClC-5 rWT (rWT) cells treated with vehicle or 10 μM FH535. Tubulin was used as a loading control in the lysates. The bar plot shows the quantification of MUC1 intracellular levels in control (Ctrl, black dots), *CLCN5* KD (KD, pink dots), and ClC-5 rWT (rWT, green dots) cells treated with vehicle or 10 μM FH535 (N ≥ 3). **(E)** MUC1 protein levels in the lysates of control (Ctrl), *CLCN5* KD (KD), and ClC-5 rWT (rWT) cells treated with vehicle or 1.5 ng/ml TGF-β. Tubulin was used as a loading control in the lysates. The bar plot shows the quantification of MUC1 intracellular levels in control (Ctrl, black dots), *CLCN5* KD (KD, pink dots), and ClC-5 rWT (rWT, green dots) cells treated with vehicle or 1.5 ng/ml TGF-β (N ≥ 3). **(F)** MUC1 protein levels in the lysates of control (Ctrl), *CLCN5* KD (KD), and ClC-5 rWT (rWT) cells treated with vehicle or 100 nM bafilomycin. Tubulin was used as a loading control in the lysates. The bar plot shows the quantification of MUC1 intracellular levels in control (Ctrl, black dots), *CLCN5* KD (KD, pink dots), and ClC-5 rWT (rWT, green dots) cells treated with vehicle or 1.5 ng/ml TGF-β (N ≥ 3). Abbreviations: MUC1, mucin-1; Ctrl, control cells; KD, *CLCN5* KD cells; rWT, ClC-5 rWT cells; BAF, bafilomycin. Statistical significance was determined using one-way ANOVA followed by Tukey's post hoc test. *$P < 0.05$, **$P < 0.01$, ***$P < 0.001$, ****$P < 0.0001$.
Source data are available for this figure.

levels are highly dependent on recycling cycles that ensure correct O-glycosylation and increase its stability at the PM (35, 36). Thus, because ClC-5 regulates endolysosomal acidification, one possibility is that lack of ClC-5 disrupts MUC1 recycling leading to a reduction of its levels. To test this possibility, we analysed MUC1 levels in cells treated with bafilomycin. Our data

showed that MUC1 levels are strongly reduced in control cells treated with bafilomycin (55% reduction, $P < 0.001$) (Fig 6F). Therefore, we suggest that impairment of endolysosomal acidification leads to MUC1 level reduction, which releases β-catenin from the PM and allows its translocation to the nucleus, promoting cell dedifferentiation and collagen transcription.

## Mice lacking *Clcn5* present a renal fibrotic phenotype with increased Col IV deposition

Finally, we decided to confirm our results in vivo using the *Clcn5* knockout mice (*Clcn5*$^{-/-}$ mice), which reproduce human DD1 pathophysiology, presenting low molecular weight proteinuria and reduced uptake of filtered proteins at PTC ([16], [17]). However, the study of fibrosis in these mice has not been addressed. First, we proceeded to evaluate it in kidneys from *Clcn5*$^{+/+}$, *Clcn5*$^{+/-}$, *Clcn5*$^{-/-}$ mice. To facilitate the analysis and taking into account that, contrarily to what happens in humans, female heterozygous mice also present the DD1 phenotype ([37]), *Clcn5*$^{+/-}$ and *Clcn5*$^{-/-}$ mice were grouped together. Haematoxylin and eosin (H/E) staining showed no obvious histological differences in glomerulus structure, tubule formation, or inflammatory infiltrate between conditions ([Fig 7A], Table S2). We then stained the kidney slices with Sirius Red/Fast Green to determine the total levels of collagens. This staining revealed that *Clcn5*$^{+/-}$ and *Clcn5*$^{-/-}$ mice present a thicker basement membrane than *Clcn5*$^{+/+}$ mice (1.39 µm versus 0.76 µm, $P < 0.05$) ([Fig 7A and B]). Specific staining of collagen type IV (Col IV), which is increased in renal fibrosis, revealed higher extracellular deposition of Col IV in mice lacking *Clcn5* (*Clcn5*$^{+/-}$ and *Clcn5*$^{-/-}$) compared with *Clcn5*$^{+/+}$ mice (1.14 µm versus 0.81 µm, $P < 0.01$) ([Fig 7A and C]), reproducing the results from our *CLCN5* KD cells. Furthermore, *Col4a1* (Col IV) mRNA levels were also increased in kidneys from *Clcn5*$^{+/-}$ and *Clcn5*$^{-/-}$ mice (1.6-fold increase, $P < 0.05$) ([Fig 7D]). Finally, global assessment considering Sirius Red/Fast Green and Col IV staining, amongst others (Table S2), confirmed that 3-mo-old *Clcn5*$^{+/-}$ and *Clcn5*$^{-/-}$ mice already present higher levels of renal fibrosis than *Clcn5*$^{+/+}$ mice (no fibrosis: 100% versus 33.3%, fibrosis degree >1: 66.6% versus 0%, $P < 0.01$) ([Fig 7E]).

## Discussion

Mutations in *CLCN5*, the genetic cause of Dent's disease type 1 (DD1), lead to PTC epithelial cell dedifferentiation and dysfunction ([8], [18]). Importantly, most DD1 patients will develop renal fibrosis with age and progress to renal failure ([20]). Renal fibrosis is characterized by pathological accumulation of ECM in the renal interstitium, which is mainly composed of collagens, being collagen I (Col I) and collagen IV (Col IV) specifically increased in renal fibrosis ([1], [2], [3]). However, how can ClC-5, a Cl$^-$/H$^+$ antiporter located at endosomes and the plasma membrane, have a role in collagens' production and release? Here, we provide a plausible novel mechanism linking ClC-5 with collagens and renal fibrosis. We propose that ClC-5 modulates Col I/IV intracellular and extracellular levels by controlling their synthesis–degradation equilibrium via the β-catenin pathway and endolysosomal acidification. In the absence of ClC-5, there is a dysregulation of this cycle that ends in collagens' overproduction and release, which could lead to renal fibrosis, mimicking what is observed in up to 60% of DD1 patients' kidney biopsies ([15]).

Using DD1 cellular models, we have shown that ClC-5 loss of function leads to a massive increase in intracellular and extracellular levels of collagen type I and type IV (Col I and IV). Notably,

we found that this phenotype is specific for collagens, because lack of ClC-5 does not increase production or release of other secretory cargoes like fibronectin 1, a component of the ECM, or cathepsin D, a lysosomal enzyme. In fact, cathepsin D is strongly down-regulated in cells depleted of ClC-5. This lysosomal aspartic protease is responsible for protein degradation, and it is significantly expressed in PTC epithelial cells. Cathepsin D has an important cytoprotective role after ischaemia/reperfusion injury, and its deficiency sensitizes RPTECs to this damage ([38]). Therefore, the decrease of cathepsin D levels in ClC-5–deleted cells may also contribute to the tubular dysfunction observed in DD patients and halt the recovery after damage, thereby further promoting renal fibrosis.

We deeply studied the role of ClC-5 in this increase in collagens' production. Our findings revealed that this is the consequence of enhanced synthesis and impaired degradation. First, we have found that Col I/IV are mostly degraded lysosomally in RPTEC/TERT1 cells, a process that is altered in *CLCN5* KD cells. In fact, and contrarily to mainstream dogma ([8]), our results show that *CLCN5* depletion does affect the lysosomal pH. One possibility we cannot fully discard is that by affecting the endolysosomal system pH, deletion of *CLCN5* is not directly affecting collagens' degradation, but their targeting to lysosomes. Importantly, collagen I and IV are big molecules with little flexibility that do not fit classic COP-II vesicles and need bigger carriers to be transported out of the ER, a process that requires TANGO1 ([39]). It has been shown that a subset of procollagen molecules are transported directly to the lysosomes from the ER for degradation to remove excess collagen from cells ([40]). Our results, indeed, also support this as a significant percentage of total collagens are located in lysosomes of *CLCN5* KD, which are bigger and closer to the ER. Moreover, we have found that inhibition of lysosomal acidification by bafilomycin A1 increases localization of Col I and IV at lysosomes and decreases their colocalization with Golgi markers, supporting our hypothesis that defects in lysosomal pH led to collagens' accumulation in these organelles. Further analysis of lysosomes' distribution revealed a higher concentration at the perinuclear area in ClC-5–depleted cells. Although less acidic lysosomes have been reported to be more peripheral ([41]), in our case we suggest that this increase in lysosomes with higher pH at the perinuclear region of CLCN5 KD cells is the reflection of impaired collagens' degradation. Collagens that are transported from the ER to the nearby lysosomes for degradation cannot be removed and accumulate inside. In addition, the dedifferentiation process previously shown by our group ([18]) because of ClC-5 loss of function is another possibility that could explain this change in lysosomal distribution. Second, our results also revealed that Col I and IV accumulate at the endoplasmic reticulum of *CLCN5* KD cells. In agreement, we have shown that *CLCN5* KD cells present higher β-catenin nuclear translocation, which promotes collagen transcription leading to higher synthesis. Interestingly, our data have shown that levels of epithelial markers MUC1 and E-cadherin, which can anchor β-catenin at PM and prevent its function ([34], [42]), are reduced in cells lacking ClC-5. This reduction in their levels would explain enhanced β-catenin nuclear translocation that is observed in *CLCN5* KD cells.

Importantly, only the WT form of ClC-5, but not the pathogenic variants, rescued the *CLCN5* KD phenotype. In addition, and resembling phenotypic variability observed in DD1 patients ([13]), each

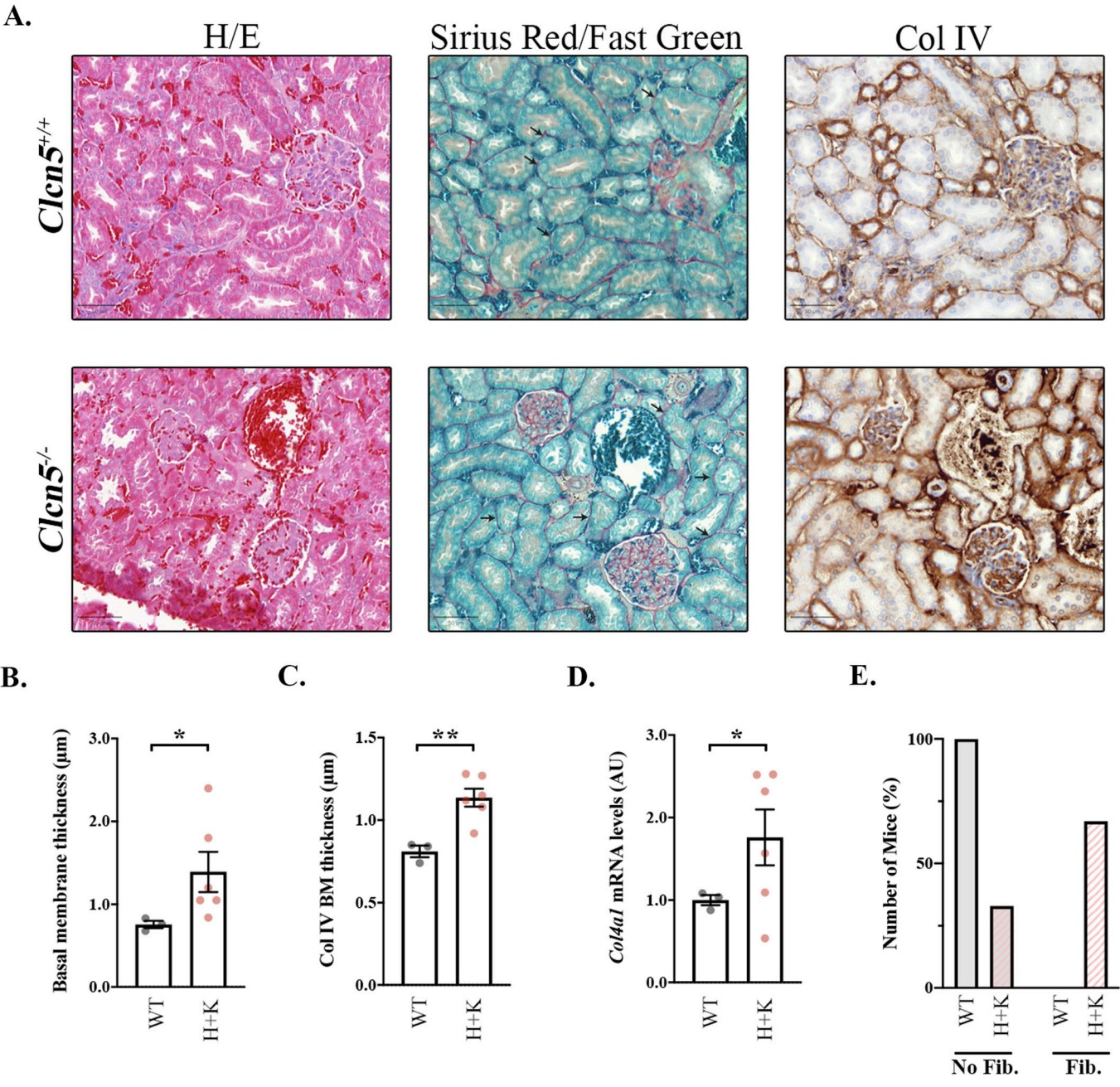

**Figure 7. Clcn5+/− and Clcn5−/− mice show increased basement membrane thickness.**
**(A)** Representative kidney slices of *Clcn5+/+* (upper panel) or *Clcn5−/−* (lower panel) mice stained with haematoxylin/eosin (H/E), with Sirius Red/Fast Green (red: all collagens, green: non-collagenous proteins), or against collagen type IV (brown, Col IV). Scale bar: 50 μm. Black arrows indicate the basement membrane. **(B, C)** Quantification of the basement membrane thickness in the kidneys stained with Sirius Red/Fast Green (B) or against Col IV (C) of *Clcn5+/+* (WT, grey dots) or *Clcn5+/−* and *Clcn5−/−* mice (H + K, pink dots). Average values ± SEM are plotted as a scatter plot with a bar graph. The y-axis represents the thickness of the basement membrane in μm. Statistical significance was determined using a two-tailed unpaired *t* test with Welch's correction. **(D)** *Col4a1* (Col IV) mRNA levels from kidneys of *Clcn5+/+* (WT, grey dots) or *Clcn5+/−* and *Clcn5−/−* mice (H + K, pink dots) normalized by *Hprt1* levels. Average values ± SEM are plotted as a scatter plot with a bar graph. Statistical significance was determined using a one-tailed unpaired *t* test with Welch's correction. **(E)** Percentage of *Clcn5+/+* (WT) or *Clcn5+/−* and *Clcn5−/−* mice (H + K) not presenting any evidence of fibrosis (No Fib.) or showing some level of fibrosis (Fib.). Abbreviations: H/E, haematoxylin/eosin staining; Col IV, collagen type IV; WT, *Clcn5+/+* mice; H + K, *Clcn5+/−* and *Clcn5−/−* mice; BM, basement membrane (N ≥ 3). *P < 0.05, **P < 0.01.

ClC-5 mutant tested here had different rescue effects on Col I/IV release, as well as on MUC1 levels and trafficking. The differential recovery of the *CLCN5* KD phenotype may correlate to the mutant expression levels, remaining ClC-5 function, or intracellular localization as previously shown (18). For instance, it is not surprising that the I524K mutant, which causes ClC-5 retention at the ER leading to their degradation (18), presented a phenotype more similar to *CLCN5* KD. On the contrary, the V523del mutant that can traffic further intracellularly and presents complex glycosylation is more similar to the WT form. Interestingly, although the mutants

## A. CONTROL SITUATION

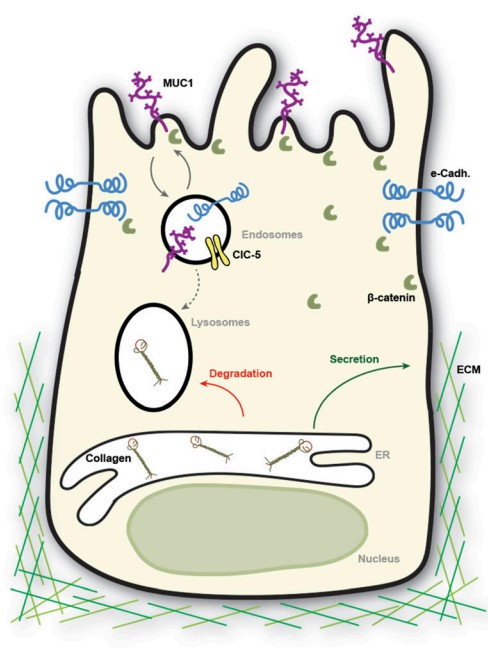

## B. CELLS LACKING ClC-5

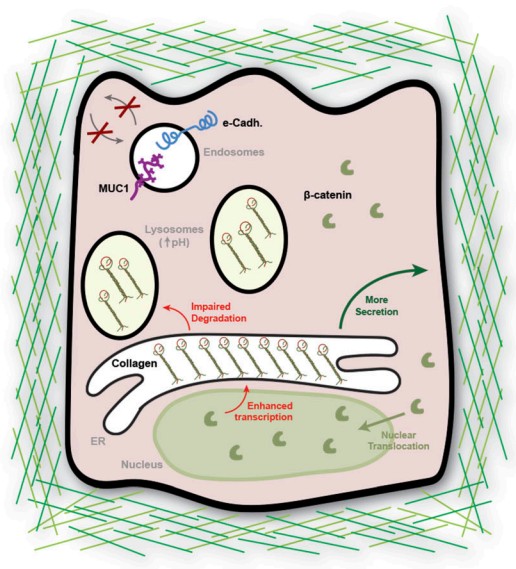

**Figure 8. Model for ClC-5 role in renal fibrosis in DD1.**
**(A)** In normal situations, ClC-5 maintains the trafficking of epithelial markers to PM, which sequesters β-catenin and ensures the correct polarization and differentiation of the renal proximal tubule epithelial cells. In addition, ClC-5 facilitates acidification of the endolysosomal pathway, allowing the degradation of bad-quality collagens. **(B)** In the absence of ClC-5, cellular dedifferentiation is induced, possibly beginning by a decrease in MUC1/E-cadherin plasma membrane levels, which will release β-catenin that will be translocated to the nucleus and promote Col I/IV transcription. Consequently, collagen synthesis is increased and, together with impaired lysosomal degradation because of acidification defects, ends in massive intracellular accumulation of Col I/IV. Next, cells will steadily release the newly synthesized collagens to the external medium, also increasing their extracellular levels that may lead to renal fibrosis over time. Abbreviations: e-Cadh., E-cadherin; ECM, extracellular matrix; ER, endoplasmic reticulum.

cannot fully restore MUC1 levels to control situation, their effect on collagens is more variable. A possible explanation is that even low levels of MUC1 are sufficient to sequester ß-catenin at the plasma membrane, therefore preventing enhanced transcription of Col I/IV. Altogether, our results are a first step to understand renal fibrosis development in DD1 patients, and future studies on different ClC-5 mutants are needed to better assess their contribution.

Finally, it is satisfying to note that the effects of *CLCN5* loss on collagens' production by RPTEC/TERT1 cell lines are replicated in the kidneys of *Clcn5* knockout (KO) mice. Our data showed that *Clcn5* KO (*Clcn5*[−/−]) and heterozygous (*Clcn5*[+/−]) mice already present evidence of renal fibrosis at 3 mo of age, reproducing this DD1 feature. One of the main findings was the thickening of the basement membranes in *Clcn5*[+/] and *Clcn5*[−/−] mice, which was also clearly observed by specific Col IV staining, a major element of renal ECM. Accordingly, other authors have shown that ClC-5 over-expression in unilateral ureteral occlusion mice causes a decrease of Col IV levels and ameliorates UOO-induced renal injury (43). Future research involving aged mice would strengthen these results as renal fibrosis in DD1 patients is a progressive process and is not normally observed during childhood (20). Besides, further investigation using patient-derived organoids could provide more information regarding the involvement of ClC-5 in renal fibrosis than cell culture systems. Another option would be to develop

coculture systems involving renal PTC cells, endothelial cells, and stromal cells to provide the proper microenvironment and ECM production (44).

To sum up, we have demonstrated in vitro and in vivo that lack of ClC-5, the genetic cause of DD1, causes a massive increase in the production and release of collagen types I and IV by altering the synthesis–degradation equilibrium. On the one hand, ClC-5 loss impairs collagens' lysosomal degradation, and on the other hand, it increases their synthesis by promoting β-catenin nuclear translocation that ends in enhanced release.

In conclusion, our data indicate that normal RPTEC/TERT1 cells tightly control collagens' production and release, to prevent dysregulated accumulation that could be detrimental to renal function and lead to renal fibrosis. Based on our results, we propose a model where ClC-5 is a key regulator of collagens' synthesis and secretion by modulating the acidification of the endolysosomal system, which impacts β-catenin pathway activation and lysosomal degradation (Fig 8A and B). To sum up, our data provide a new mechanism for ClC-5 role in renal fibrosis by regulating Col I/IV levels. In addition, our results may be also relevant for other renal Fanconi syndrome–related diseases also progressing to renal fibrosis. Finally, we suggest that modulators of these pathways could be used as therapeutic strategies to prevent uncontrolled collagen release and, ultimately, halt Dent's disease type 1 progression.

# Materials and Methods

## Cell culture

RPTEC/TERT1 (#CRL-4031; ATCC) cells were grown in DMEM: Nutrient Mixture F-12 (1:1, vol/vol) (#31331093; Thermo Fisher Scientific) supplemented with 20 mM Hepes (#15630–080; Gibco), 60 nM sodium selenite (#S9133; Sigma-Aldrich), 5 $\mu$g/ml transferrin (#T1428; Sigma-Aldrich), 50 nM dexamethasone (#D8893; Sigma-Aldrich), 100 U/ml penicillin and 100 $\mu$g/ml streptomycin (#15240–062; Gibco), 2% FBS (#10270; Gibco), 5 $\mu$g/ml insulin (#I9278; Sigma-Aldrich), 10 ng/ml EGF (#E4127; Sigma-Aldrich), and 3 nM triiodothyronine (#T5516; Sigma-Aldrich). Culture conditions were maintained at 37°C in a humidified atmosphere containing 5% $CO_2$. RPTEC/TERT1 control cells, *CLCN5* knockdown cells (*CLCN5* KD), and knockdown cells overexpressing the ClC-5 WT form (ClC-5 rWT) or the different mutant forms (ClC-5 rV5223del, ClC-5 rE527D, and ClC-5 rI524K) were obtained by stable transduction as described in reference 18.

## Mycoplasma test

Mycoplasma contamination was analysed by PCR. The media of the RPTEC/TERT1 cell line were collected and centrifuged at 300$g$ for 3 min to remove cell debris. The supernatant was centrifuged at 15,000$g$ for 10 min to sediment the mycoplasma. The supernatant was carefully discarded, and the pellet was resuspended in Milli-Q water. Samples were heated at 95°C for 5 min. Then, PCR amplification of mycoplasma DNA was performed as described in reference 18. Briefly, two $\mu$l of DNA was amplified in a 25 $\mu$l final volume of reaction mix containing 2.5 mM $MgCl_2$, 0.2 mM dNTPs, 0.2 $\mu$M forward primer mix (5′-CGCCTGAGTAGTACGTWCGC-3′, 5′-TGCCTGRGTAGTACATTCGC-3′, 5′-CRCCTGAGTAGTATGCTCGC-3′, 5-′CGCCTGGGTAGTACATTCGC-3′), 0.2 $\mu$M reverse primer mix (5′-GCGGTGTGTACAARACCCGA-3′, 5′-GCGGTGTGTACAAACCCCGA-3′), and 1 unit of Taq polymerase (Bioline, Meridian Bioscience). The PCR conditions were an initial melting step of 94°C for 4 min; then 35 cycles of 94°C for 30 s, an annealing step of 55°C for 30 s, and 72°C for 45 s; followed by a final elongation step of 72°C for 6 min. PCR products were separated in 2% agarose gels.

## *Clcn5*$^{+/+}$, *Clcn5*$^{+/-}$, and *Clcn5*$^{-/-}$ mice

Paraffined kidneys from *Clcn5*$^{+/+}$, *Clcn5*$^{+/-}$, and *Clcn5*$^{-/-}$ mice (N = 3 female mice for each condition, 3 mo old) were previously generated and characterized by Dr. Baisong Lu (Wake Forest Institute for Regenerative Medicine) (37). It is important to state that heterozygous mice also present the DD1 phenotype (37), so *Clcn5*$^{+/-}$ and *Clcn5*$^{-/-}$ mice were grouped together in the subsequent analyses.

## H/E, Sirius Red/Fast Green, and Col IV staining

Kidney sections were stained with H/E, Sirius Red/Fast Green Collagen, or Col IV for histological analysis as described previously (45). Sirius Red was used to stain all types of collagens, whereas

Fast Green was used to stain non-collagenous proteins. H/E-stained sections were evaluated by light microscopy using a semiquantitative analysis assessing the following parameters: epithelial hyperplasia (N, normal; 1, minimal; 2, mild; 3, moderate; 4, marked), inflammatory infiltrate in the lamina propria (N, normal; 1, minimal; 2, mild; 3, moderate; 4, marked), oedema of the lamina propria (0, normal; 1, minimal; 2, mild; 3, moderate; 4, marked), structure glomeruli (N, normal; A, altered), and structure tubules (N, normal; A, altered). The histological study was performed in a blinded fashion.

## Measurement of basement membrane thickness

The thickness of the basement membrane was measured in kidney tissue sections using the ruler tool of QuPath software (46). Ten different measures were performed per stain (Sirius Red/Fast Green and Col IV) of each mouse.

## Collagen secretion assay and Western blot

The media of RPTEC/TERT1 cells were replaced with fresh medium for up to 24 h to perform a collagen secretion assay. The media were collected and centrifuged at 1,500$g$ to remove cell debris. The supernatants (SN) were denatured at 95°C for 5 min with Laemmli SDS sample buffer. For cell extracts, cells were washed with PBS 1x and lysed with SET 1x buffer (10 mM Tris–HCl, pH 7.4, 150 mM NaCl, 1 mM EDTA, and 1% SDS). The cell extracts and SN were resolved in 15% SDS–PAGE and transferred to PVDF membranes (#ISEQ00010; Millipore) at 100 V during 3 h. Membranes were blocked with non-fat dry milk diluted in PBS-T (1x PBS, 0.1% Tween-20) for 1 h and incubated overnight at 4°C with collagen I (#ab34710; Abcam), collagen IV (#ab6586; Abcam), MUC1 (MA1-06503; Thermo Fisher Scientific), E-Cadherin (610181; BD Transduction Labs), LAMP-1 (#53-1079-42), ClC-5 (#GTX53963; Gentex), fibronectin 1 (#ab45688; Abcam), cathepsin D (EPR3057Y; Abcam), and tubulin (T4026; Sigma-Aldrich) antibodies. Band intensities were visualized using Odyssey Fc Imaging System (LI-COR) and analysed using FIJI.

## Collagen degradation assay

For collagen degradation assay, RPTEC/TERT1 cells were cultured into plates for 10 d to allow cell differentiation. Intracellular and extracellular levels of collagen I and IV after treatment with vehicle (DMSO), proteasome inhibitor (MG132; # 474790; Sigma-Aldrich), bafilomycin A1 (#19-148; Sigma-Aldrich), or lysosomal enzymatic activity inhibitor (leupeptin; # L2884; Sigma-Aldrich) were analysed by Western blot as described previously.

## Cycloheximide chase assay

The turnover rates of Col I and IV were determined by cycloheximide (#C7698; Sigma-Aldrich) chase assays. RPTEC/TERT1 cells were incubated with 50 $\mu$g/ml CHX or DMSO (vehicle) and collected at different times (0, 3, 6, 8, 24 h). Then, cellular extracts were analysed by Western blot as described previously. Quantification of collagen type I and IV half-life was carried out using the

non-linear regression one-phase exponential decay model of GraphPad Prism 9 software (RRID:SCR_002798) for each independent experiment (N ≥ 3).

## Microscopy and immunofluorescence

RPTEC/TERT1 cells were cultured on coverslips during 7 d. Cells were fixed with 4% PFA for 20 min at RT. Aldehyde groups were quenched in 50 mM $NH_4Cl$/PBS for 30 min, and non-specific binding sites were blocked with 5% BSA in PBS for 60 min. Primary antibodies were diluted in blocking reagent and incubated overnight at 4°C (collagen I [# ab34710; Abcam], collagen IV [#ab6586; Abcam], MUC1 [MA1-06503; Thermo Fisher Scientific], LAMP-1 [#53-1079-72; Invitrogen], KDEL [#Ab10C3; Abcam], $\beta$-catenin [#ab32572; Abcam]). Secondary antibodies conjugated with 488 (#A28175; Invitrogen) and 568 (#A28175; Invitrogen) were diluted in a blocking solution and incubated for 1 h at RT. Cells were incubated with Hoechst 33342 (1:2,000 dilution, #H1399; Invitrogen) or phalloidin (1:200 dilution, #R415; Invitrogen) for 1 h at RT. Fluorescence was visualized in a confocal spectral Zeiss LSM 980 microscope. Images were processing using FIJI (47). Manders' overlap coefficient was calculated using the ImageJ plugin JACoP (48) in a minimum of three different fields (>5 cells/field) for three independent experiments per condition. JACoP plugin measures colocalization using different indicators (Pearson's coefficient, overlap coefficient, or Manders' coefficients) on two images for all z-stacks.

To determine the number, volume, and area of LAMP1-, Col I–, and Col IV–positive elements, we used the 3D objects counter v2.0 tool from FIJI (48). All images for each experiment were taken on the same day under the same conditions and the same z-step (0.5 $\mu$m), and a minimum of three fields for each experiment (N > 3) were analysed. The threshold was automatically set for each image. DAPI was used to count the number of nuclei per field. Regarding quantification of collagen fibres, Col I fibres were defined as those particles with an area > 2 $\mu m^2$, perimeter > 10 $\mu$m, whereas Col IV fibres were defined as those Col IV particles with an area > 1.5 $\mu m^2$, perimeter > 10 $\mu$m.

## Real-time quantitative RT–PCR

For gene expression analysis, total RNA was isolated from RPTEC/TERT1 cells using TRIzol reagent following the manufacturer's instructions. Equal amounts of RNA (1 $\mu$g) were retro-transcribed to cDNA using High Capacity RNA-to-cDNA Kit according to the manufacturer's instructions. Gene expression was evaluated by real-time quantitative PCR using SYBR Green Master Mix (#A25742; Applied Biosystems) or TaqMan Master Mix (#4369016; Applied Biosystems), according to the manufacturer's instructions. Total levels of *CLCN5*, collagen I and IV (*COL1A1* and *COL4A1*, respectively), *SOX2*, *SOX4*, *EFNB1*, and *TBP* were evaluated with specific primers (*CLCN5* primers: 5′-GGGATAGGCACCGAGAGAT-3′ and 5′-GGTTAAACCAGAATCCCCCTGT-3′; *COL1A1* primers: 5′-GTGGTCAGGCTGGTGTGATG-3′ and 5′-CAGGGAGACCCTGGAATCCG-3′; *COL4A1* primers: 5′-ATGGGGCCCCGGCTCAGC and 5′-ATCCTCTTT-CACCTTTCAATAGC; *SOX2* primers: 5′-GAGCTTTGCAGGAAGTTTGC-3′ and 5′-GCAAGAAGCCTCTCCTTGAA-3′; *SOX9* primers: 5′-

GTACCCGCACTTGCACAAC-3′ and 5′-CGCTCTCGTTCAGAAGTCTC-3′; *EFNB1* primers: 5′-GTATCCTGGAGCTCCCTCAACC-3′ and 5′-GCTTGTAGTACTCATAGGGCCG-3′; *TBP* primers: 5′-CGGCTGTTTAACTTCGCTTC-3′ and 5′-CAGACGCCAAGAAACAGTGA-3′). In contrast, total levels of BMP4 were measured with a commercial TaqMan probe (Hs00370078_m1). Data were normalized with *TBP* (Hs00427620_m1). Amplification protocol was performed using LightCycler 480 System (Roche). Relative expression fold change was determined by the comparative $2^{(-\Delta\Delta CT)}$ method after normalizing to *TBP*.

## Endolysosomal pH determination

Endosomal pH was determined using the LysoSensor Yellow/Blue DND-160 dye (L7545; Thermo Fisher Scientific). RPTEC/TERT1 cells were cultured in IBIDI chambers for 10 d. Cells were washed three times with isotonic solution containing 2.5 mM KCl, 140 mM NaCl, 1.2 mM $CaCl_2$, 0.5 mM $MgCl_2$, 5 mM glucose, and 10 mM Hepes (305 mosmol/litre, pH 7.4 adjusted with Tris), and then treated with LysoSensor for 10 min at 37°C. After the incubation period, cells were washed again with isotonic solution and imaged using Leica Thunder Imager 3D Cell Culture Microscope for 3 min without stimulation. In continuation, cells were treated with vehicle (DMSO) or 100 nM bafilomycin A1 to get a normalization point. Images were acquired with a 5X objective on Leica Thunder Imager 3D Cell Culture Microscope and processed using ImageJ software.

## Cathepsin B activity measurement

The Magic Red Cathepsin B Kit (#ICT938; Bio-Rad) was used to quantify and monitor cathepsin B activity in RPTEC/TERT1 cell lines. Briefly, differentiated control, *CLCN5* KD, and ClC-5 rWT RPTECs were treated with either DMSO (vehicle) or bafilomycin A1, as a positive control, for 24 h. Next day, cells were washed three times with isotonic solution containing 2.5 mM KCl, 140 mM NaCl, 1.2 mM $CaCl_2$, 0.5 mM $MgCl_2$, 5 mM glucose, and 10 mM Hepes (305 mosmol/litre, pH 7.4 adjusted with Tris), and incubated with Magic Red cathepsin B reactive at 1:26 (vol/vol) for 5 min at 37°C (following the manufacturer's protocol). After the incubation period, the cells were imaged using Leica Thunder Imager 3D Cell Culture Microscope. All data were processed and analysed using FIJI (47).

## *β*-Catenin nuclear translocation

Nuclear translocation of $\beta$-catenin was analysed by immunofluorescence. Briefly, cell lines were cultured on IBIDI chambers during 10 d and, after this period, starved overnight and exposed to 1.5 ng/ml of TGF-$\beta$ (#100-B-001; R&D Systems) or 10 $\mu$M FH535 (TO-4344/10 MG; Biogen) for 24 h. At the end of treatment, cells were washed and lysed or fixed as described previously. Fluorescence was visualized in a confocal spectral Zeiss LSM 980 microscope. Images were processed and analysed using FIJI (47).

## Subcellular fractionation assay

Subcellular fractionation was performed using Subcellular Protein Fractionation Kit (#78840; Thermo Fisher Scientific) following the manufacturer's instructions. Differentiated RPTEC/TERT1 cell lines were treated with vehicle (DMSO) or TGF-$\beta$ (1.5 ng/ml) during 24 h. At the end of treatment, cells were washed with PBS, lysed, and incubated with different buffers to recover different subcellular fractions (plasma membrane, cytosol, and nuclear extract) according to the manufacturer's instructions. Protein concentration was quantified by the BCA protein kit. Samples were heat-denatured in Laemmli buffer and subjected to SDS–polyacrylamide gel electrophoresis.

## Statistical analysis

All data are means ± SEM. In all cases, a D'Agostino–Pearson omnibus normality test was performed before any hypothesis contrast test. Statistical analysis and graphics were performed using GraphPad Prism 9 (RRID:SCR_002798) software. For data that followed normal distributions, we applied either a $t$ test or one-way ANOVA followed by Tukey's post hoc test. For data that did not fit a normal distribution, we used Mann–Whitney's unpaired $t$ test and non-parametric ANOVA (Kruskal–Wallis) followed by Dunn's post hoc test. Binary and categorical variables were analysed by Fisher's exact test. Criteria for a significant statistical difference were as follows: $*P < 0.05$, $**P < 0.01$, $***P < 0.001$, $****P < 0.0001$.

# Data Availability

The authors confirm that the data supporting the findings of this study are available within the article or its supplementary materials. Correspondence and material requests should be addressed to G Cantero-Recasens (Gerard.cantero@vhir.org) or A Meseguer (ana.meseguer@vhir.org).

# Supplementary Information

# Acknowledgements

We thank the patient advocacy group ASDENT (Asociación de la Enfermedad de Dent, www.asdent.es) for continuously supporting our group, and also the Dent Disease Foundation (DD Foundation, www.dentdisease.org) for promoting international cooperation in Dent Disease Research. We also thank all members of the Renal Physiopathology Group for valuable discussions, especially C García for technical support. Specific primers to detect mouse *Col4a1* and *Hprt1* were kindly provided by Dr. Jacobs. Fluorescence microscopy was performed at the High Technology Unit (UAT) at the Vall d'Hebron Research Institute (VHIR). Mouse tissue staining was performed at the Translational Molecular Pathology Unit (VHIR), and we thank T Moliné for her assistance. This work reflects only the authors' views, and the EU Community is not liable for any use that may be made of the information contained therein. This work was supported in part by Dent's Disease Patients Association ASDENT and grants from Ministerio de Ciencia e Innovación (SAF201459945-R and SAF201789989-R to A Meseguer), Instituto de Salud Carlos III (PI22/00741 to G Cantero-Recasens and A Meseguer, cofinanced by the European Union), the Fundación Senefro (SEN2019 to A Meseguer), and the Mizutani Foundation (REF230040 to G Cantero-Recasens). A Meseguer research group holds the Quality Mention from the Generalitat de Catalunya (Grant No. 2021 SGR 01600).

## Author Contributions

M Durán: conceptualization, formal analysis, validation, investigation, methodology, and writing—review and editing.
G Ariceta: investigation and writing—review and editing.
ME Semidey: formal analysis and writing—review and editing.
C Castells-Esteve: investigation and writing—review and editing.
A Casal-Pardo: investigation and writing—review and editing.
B Lu: investigation and writing—review and editing.
A Meseguer: conceptualization, resources, formal analysis, supervision, funding acquisition, and writing—review and editing.
G Cantero-Recasens: conceptualization, formal analysis, supervision, funding acquisition, validation, investigation, methodology, and writing—original draft, review, and editing.

## Conflict of Interest Statement

The authors declare that they have no conflict of interest.

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
