## [Reviewer comments · Life Science Alliance]

Life Science Alliance

Renal antiporter CIC-5 regulates collagen I/IV through β -catenin pathway and lysosomal degradation

Mónica Durán, Gema Ariceta, Maria Eugenia Semidey, Carla Castells-Esteve, Andrea Casal-Pardo, Baisong Lu, Anna Meseguer, and Gerard Cantero-Recasens

DOI: <https://doi.org/10.26508/lsa.202302444>

Corresponding author(s): Gerard Cantero-Recasens, Vall d'Hebron Institut de Recerca and Anna Meseguer, Vall d'Hebron Research Institute-CIBBIM Nanomedicine

Review Timeline:

Submission Date:	2023-10-18
Editorial Decision:	2023-11-24
Revision Received:	2024-03-25
Editorial Decision:	2024-04-09
Revision Received:	2024-04-15
Accepted:	2024-04-17

Transaction Report:

November 24, 2023

Re: Life Science Alliance manuscript #LSA-2023-02444-T

Dr. Gerard Cantero-Recasens
Vall d'Hebron Institut de Recerca
Renal Physiopathology Group
Pg. Vall d'Hebron 119-129
Barcelona 08035
Spain

Dear Dr. Cantero-Recasens,

Thank you for submitting your manuscript entitled "Renal Cl-/H+ antiporter CIC-5 regulates collagen I/IV levels through β -catenin pathway and lysosomal degradation" to Life Science Alliance. The manuscript was assessed by expert reviewers, whose comments are appended to this letter. We invite you to submit a revised manuscript addressing the Reviewer comments.

Thank you for this interesting contribution to Life Science Alliance. We are looking forward to receiving your revised manuscript.

Sincerely,

B. MANUSCRIPT ORGANIZATION AND FORMATTING:

Reviewer #1 (Comments to the Authors (Required)):

Duran et al describe the role of CIC-5, the genetic cause of Dent's disease, on col I and IV expression. Using cell-based approaches and in vivo studies, they demonstrated that upon CIC-5 knockdown, the acidification of the endolysosomal system is altered, impacting beta-catenin activation and lysosomal degradation. The consequence is a massive increase in the production and release of collagen I and IV by altering the equilibrium synthesis-degradation.

This study provides a new mechanism for CIC-5 role on renal fibrosis and potential interesting perspectives for new therapeutic strategies to prevent excessive collagen expression and release, which could ultimately prevent Dent's disease and renal fibrosis progression.

Overall, the manuscript is well written and the study well designed. However, some of the authors' conclusions need to be strengthened by additional experiments or more rigorous analysis at several points in the paper.

1- One key point that needs to be addressed is whether the phenotype observed in CIC-5 KD cells is specific on Col I and IV expression and secretion, or whether it affects other secreted proteins. Secretome analysis or biotinylated cell surface protein analysis, or at least, investigate the secretion of other individual cargo proteins should be performed.

2- Supplementary figure 2A, the authors state in lines 238-239 that "These mutants were correctly expressed in respective cell lines (previously generated by our lab (18)) as assessed by WB". However, as shown in the corresponding figure, the expression of the different pathogenic CIC-5 mutants (V523del, E527D and I524K) appears very low compared to the expression of CIC-5 rWT. Thanks for your comments.

3- Supplementary figure 2B and 2C. Is the quantification indicated from only a single experiment, or were other experiments performed here? Please, specify in the figure legend. Quantification from at least 3 independent experiments should be shown. Moreover, from the image shown, the reviewer disagree with the conclusion made by the authors lines 239 to 241: "our data showed that loss-of-function E527D and I524K mutants did not rescue CLCN5 KD phenotype on Col I/IV intracellular and extracellular levels. Expression of V523del mutant, however, did partially restore both Col I / IV intracellular and extracellular levels to those of rWT condition." Indeed, on the basis of the quantification, it appears that for all the CIC-5 mutants the level of Col I/IV remains elevated compared CIC-5 rWT. It would be more convincing to also include in these westernblotting experiments, the condition shown in figure 1A and 1C (ctr and KD).

4- Figure 1 E. Confocal analysis of Coll and IV expression must be quantified by specifying in the figure legend or in the material and methods sections the number of experiments and the number of cells analyzed for quantification.

5- Figure 1G. Please provide details of quantification (number of fields or cells quantified? How did you identified individual collagen fibers ?...).

6- Figure 2A and 2B, could the author explain why they used KDEL as an ER marker, and not a more conventional marker such as calnexin? And wouldn't it be more rigorous to analysis the Manders's coefficient specifically for 40-50 individual cells per condition? The figure legend does not mention the number of fields or cells taken into account for the quantification shown in figures 2C and 2F.

7- Figure 3C and 3D, once again, how many experiments were performed? This is not specified in the figure legend. It is difficult for the reviewer to be convinced based on the difference in tubulin expression. Especially in figure 3C for the line 3 for the blot ColIV, and on figure 3D, lanes 3 and 4 for the lysate samples. In addition, the signal for col I in figure 3C is very low in lane 1 and 2 (here, a membrane with a higher exposure should be shown).

8- On the basis of the data presented in figure 3, the reviewer believe that at this stage, there is an over-interpretation of the result when the authors state lines 287 and 288 that "These data strongly suggest that CIC-5 impairs collagens degradation by altering lysosomal acidification.

9- To reinforce the results suggesting a defect in lysosome degradative activity upon CLC-5 KD, it is important to use other specific assay to test whether the degradation of other substrates is also compromised (ex Magic red assay, or cathepsin L or D activity assay...)

10- Figure 4C, what is the rationale for assessing the colocalization of LAMP1 with KDEL? In terms of lysosome distribution and intracellular location, lysosomes are generally known to be more concentrated in the perinuclear area, and it has been reported that less acidic lysosomes are more peripheral. So, taking into account the effect of CLC-5 KD on lysosomal pH as shown in figure 4A, the quantification of perinuclear and peripheral lysosomes would be more relevant than the colocalization with KDEL.

11- Line 299, authors indicate that the number of lysosomes was studied, but there are no results in figure 4 showing this analysis.

12- Figure 5A, the effect of TGFbeta on col 1 and 4 mRNA expression is only tested in control cells. However, it is essential to test whether TGFbeta also increases col 1 and 4 mRNA expression in CLC-5 depleted cells in order to assess if TGFbeta and CLC-5 are involved in the same pathway (as it is done on figure 5D for protein expression).

13- The immunofluorescence results in figure 5B suggesting that upon TGF beta treatment or CLC-5 KD, beta-catenin is translocated from the plasma membrane to the nucleus should be reinforced by a biochemical approach using cytoplasmic and nuclear fractions from which beta-catenin expression is quantified by western blotting.

14- Figure 6B, 6E and 6F, the loading control (tubulin) is extremely variable between the different conditions tested. It would be advisable to use another loading control such as actin or GAPDH to ensure that the same quantity of protein extract is loaded.

Reviewer #2 (Comments to the Authors (Required)):

The authors present an interesting manuscript describing the potential role of CLC-5 in controlling cellular collagen homeostasis. The data suggest that CLC-5 functions in acidifying the endolysosomal system and a loss of function results in two complementary changes that affect the levels of collagen types I and IV. First, the trafficking of cell surface mucin 1 is affected, which results in altered signalling. Beta catenin migrates more to the nucleus, leading to increased transcription of collagens. Second, the actual degradation of collagen is reduced, since the lysosomes do not acidify appropriately. The data and model are interesting and would be of considerable interest to several researchers in diverse fields of cell biology and physiology.

I have a few minor suggestions or concerns.

1. In the first figure, it is not clear if cycloheximide was used in the experiments to assay secretion. Please repeat secretion and degradation time courses in the presence of cycloheximide to remove the confounds from altered transcription and translation that the authors have demonstrated later. These data will go toward addressing another issue that might be exciting, which is whether the CLC-5 affects the actual processes of secretion - could it actually increase the secretory capacity of the cell to allow for the increased collagen?
2. It is unclear what the 'relative secreted levels of collagen' are in figure 1D. Is the secreted collagen normalised to account for the number of cells? A way to plot it could be to normalise the secreted collagen to the tubulin. Is this what has been done? Please clarify in the figure legend. Avoid plotting a ratio of secreted collagen to total collagen - this makes interpretation confusing, since both intracellular retention and secretion are differentially affected in these cells.
3. The differential effects of the loss-of-function mutants are interesting, but they raise some questions. E527D and I524K mutants express much more poorly than the WT protein. Could this reduced expression explain the observation that they did not rescue CLCN5 KD phenotype on Col I/IV intracellular and extracellular levels? Please repeat the experiment using far less CLCN5 WT DNA to induce equivalent low expression of wild type protein for a comparable experiment.
4. Previous results from the same authors have shown that E527D and V523del have similar intracellular distributions, while only the I524K was retained in the ER. This is not consistent with the current explanation for the different phenotype with V523del. Please clarify.
5. One way to clarify the suggestion that ER-retention of the CLC-5 prevents it from reaching its appropriate site of function, is to use a CLC-5 construct under the control of the RUSH system. Under baseline conditions, the protein should be retained in the ER. Once released (by addition of biotin), the protein should be transported to its correct location. Rescue of the phenotype can then be assayed with or without biotin.
6. The 'number of fibres' is a confusing depiction of the extracellular collagen. Could the authors please present the total fluorescence associated with the extracellular collagen as well?
7. The localization of collagen type I or IV to lysosomes is not clear in the images. Arrows supposedly indicating colocalization often actually point at sites where the collagen and LAMP1 signal are next to each other, but not colocalized. The difference is several hundred nanometres. Please provide better images, remove arrows toward sites of collagen / LAMP1 signal touching each other without colocalization, and highlight some key locations with a magnified inset.
8. In figure 3D, bafilomycin seems to have enhanced collagen IV degradation. What does this mean?
9. Inhibiting organelle acidification with bafilomycin has also been shown to affect secretory function and trap secretory cargoes at the Golgi apparatus or the endolysosomal network.
10. Could the authors please show using immunofluorescence (as in

figure 2D and 2E) that treatment with inhibitors of lysosomal activity actually results in an increase in colocalization of collagen and LAMP1? If the collagen is retained in the secretory pathway, this might explain why bafilomycin has minimal effect on degradation in the lysosome.

11. In section 5, TGF- β treatment increases collagen mRNA in control cells. Please repeat this in the other two cell lines (both the depletion as well as the rescue). This will be useful to look at in light of the rest of the data in the same figure.

12. The text describing supplementary figure 3 needs to be corrected - the data are in supplementary figure 4.

13. The data from transgenic mice are a nice addition. Could the authors also carry out a similar staining for mucous? PAS staining should reveal any alterations in the mucous thickness. Similarly, an immunohistochemical staining of beta catenin to compare surface and nuclear localisation as in the cell lines.

14. As a final small issue - the text is confusing on one point. The authors suggest "our data indicates that lack of CLCN5 promotes collagen synthesis through β -catenin pathway and, in combination with lysosomal degradation impairment, provokes a huge intracellular collagen accumulation that will also end in increased release of collagens to the extracellular medium." That statement (made several times) suggests that increased secretion could come from either increased synthesis or the collagen that is stuck in lysosomes. This would be very interesting as well, if it were demonstrated. From the evidence presented, it would seem clear that the collagen retained in poorly-acidified lysosomes does not get secreted (for instance supp fig 3A, Bafilomycin). The increase in secretion must come entirely from the increased synthesis. There are no other experiments testing the hypothesis that collagen retained in the lysosomes is secreted. Please rewrite to clarify this - from the abstract to the discussion. Alternatively, authors could carry out experiments to test whether the lysosomal accumulations of collagen are secreted, though that is not in the ambit of this paper.

Reviewer #3 (Comments to the Authors (Required)):

The authors present interesting data regarding the effects of CIC-5 knocked down on cellular synthesis and secretion of collagens. The implication is that this mediated renal fibrosis and eventual chronic kidney disease/kidney failure in patients with Dent disease type 1. The fibrosis aspect of the observation is reproduced in a knock out mouse model.

Overall, the data are very compelling in the cell culture experiments. Multiple experiments are presented that demonstrate abnormal lysosomal function in the CIC-5 knock down lines. Good evidence is presented to implicate the beta catenin pathway. The cellular phenotype can be rescued with wild type CIC-5 protein but not protein that harbors human pathogenic mutations.

1. For the most part, the paper relies on observations in a single line of culture cells.

It would be ideal to see this reproduced at least in certain aspects in an independent model system.

2. A lot more could be done with the animal model to demonstrate key features in the proximal tubule consistent with the salt culture model. This might do some to alleviate concerns that there is any lack of generalizability of the cell culture studies.

3. Did the authors look at any effects within the podocytes of their culture cells in order to see if similar events could be going on there to explain the focal global sclerosis that is been described in Dent disease.

4. At a minimum, it seems the paper could benefit from a good strengths and weaknesses section in the discussion to point out some of the things that might need to be verified in either additional cell culture systems, organoids, or intact animals.

March 25, 2024

RE: Duran et al. **Renal Cl-/H+ antiporter CIC-5 regulates collagen I/IV levels through β -catenin pathway and lysosomal degradation.**

Reference: LSA-2023-02444-T

Dear Dr. Eric Sawey,

We thank you and the reviewers for your critical evaluation of our findings and for giving us the opportunity to present a revised version of our manuscript. Please find below our response to reviewers' concerns (in blue and italics).

Reviewer #1

Duran et al describe the role of CIC-5, the genetic cause of Dent's disease, on col I and IV expression. Using cell-based approaches and in vivo studies, they demonstrated that upon CIC-5 knockdown, the acidification of the endolysosomal system is altered, impacting beta-catenin activation and lysosomal degradation. The consequence is a massive increase in the production and release of collagen I and IV by altering the equilibrium synthesis-degradation. This study provides a new mechanism for CIC-5 role on renal fibrosis and potential interesting perspectives for new therapeutic strategies to prevent excessive collagen expression and release, which could ultimately prevent Dent's disease and renal fibrosis progression. Overall, the manuscript is well written and the study well designed. However, some of the authors' conclusions need to be strengthened by additional experiments or more rigorous analysis at several points in the paper.

1- One key point that needs to be addressed is whether the phenotype observed in CIC-5 KD cells is specific on Col I and IV expression and secretion, or whether it affects other secreted proteins. Secretome analysis or biotinylated cell surface protein analysis, or at least, investigate the secretion of other individual cargo proteins should be performed.

*We thank the reviewer for this comment. To answer this point, we have studied the secretion of two individual cargo proteins, Fibronectin (FN1, component of the extracellular matrix) and Cathepsin D (CTSD, typical secreted lysosomal enzyme). Our results, now included in the manuscript and in **Supplementary figure 2B**, show that Fibronectin levels (intracellular or secreted) are not affected by CIC-5 levels. Cathepsin D levels, however, are strongly reduced in CIC-5 KD cells, a phenotype partially recovered by the expression of CIC-5 rWT. This fits previous microarray data showing no differences in FN1 mRNA levels between different conditions, but a reduced CTSD expression in cells lacking a functional CIC-5 (Duran et al, 2021). These new results, which are included in both Results and Discussion sections, confirm that the effect of CIC-5 KD is specific for collagen type I and type IV. Finally, we postulate that the decrease in Cathepsin D levels may be related to the impairment of lysosomal function due to CIC-5 lack of function, which opens new research lines regarding CIC-5 role on intracellular trafficking that we will study in the future.*

RESULTS

“Next, to further assess whether this phenotype of *CLCN5* KD cells is specific for Col I / IV intracellular and extracellular levels, we studied two individual cargo proteins, Fibronectin 1 (FN1, component of the extracellular matrix) and Cathepsin D (CTSD, typical lysosomal enzyme which can be released to the extracellular medium) (26, 27). Notably, our analyses showed no major effect of *CLCN5* KD on FN1 intracellular or extracellular levels. CTSD intracellular levels, however, were strongly reduced in cells depleted of CIC-5 (75% decrease), a phenotype mildly rescued by CIC-5 rWT expression. Similarly, CTSD

release to extracellular medium was totally abolished in *CLCN5* KD cells compared to control cells (93% reduction) (**Supplementary Figure 2B**). This fits previous microarray data showing no differences in *FNI* mRNA levels between control and *CLCN5* KD cells, but a strong downregulation of CTSD in cells lacking a functional CIC-5 (18). In conclusion, our results demonstrate that *CLCN5* depletion specifically affects Collagen I / IV levels and does not have a general effect on secretory cargoes.”

DISCUSSION

“Using DD1 cellular models, we have shown that CIC-5 loss-of-function leads to massive increase of intracellular and extracellular levels of collagen type I and type IV (Col I and IV). Notably, we found that this phenotype is specific for collagens, since lack of CIC-5 does not increase production or release of other secretory cargoes like Fibronectin 1, component of the ECM, or Cathepsin D, a lysosomal enzyme. In fact, Cathepsin D is strongly downregulated in cells depleted of CIC-5. This lysosomal aspartic protease is responsible of protein degradation, and it is significantly expressed in proximal tubule epithelial cells. Cathepsin D has an important cytoprotective role after ischemia-reperfusion injury (IRI), and its deficiency sensitises RPTEC cells to this damage (42). Therefore, the decrease of Cathepsin D levels in CIC-5 deleted cells may also contribute to the tubular dysfunction observed in DD patients and halt the recovery after damage, thereby further promoting renal fibrosis.”

2- Supplementary figure 2A, the authors state in lines 238-239 that "These mutants were correctly expressed in respective cell lines (previously generated by our lab (18)) as assessed by WB". However, as shown in the corresponding figure, the expression of the different pathogenic CIC-5 mutants (V523del, E527D and I524K) appears very low compared to the expression of CIC-5 rWT. Thanks for your comments.

As discussed in Duran et al, 2021, all mutants are correctly expressed at the mRNA levels, although their protein levels vary compared to CIC-5 WT form likely due to enhanced degradation. To avoid any confusion, we are now including qPCR data showing endogenous and exogenous levels of CIC-5, confirming that all mutants are correctly expressed. We also include one sentence in the text clarifying this:

“Next, we studied the effect of different pathogenic CIC-5 mutants (V523del, E527D and I524K) on *CLCN5* KD collagen phenotype. These mutants were correctly expressed at mRNA level in respective cell lines (**Supplementary Figure 1A**), although protein levels were more variable due to mutants being delivered for degradation as previously described (**Supplementary Figure 1C**)(18).”

3- Supplementary figure 2B and 2C. Is the quantification indicated from only a single experiment, or were other experiments performed here? Please, specify in the figure legend. Quantification from at least 3 independent experiments should be shown. Moreover, from the image shown, the reviewer disagree with the conclusion made by the authors lines 239 to 241:"our data showed that loss-of-function E527D and I524K mutants did not rescue *CLCN5* KD phenotype on Col I/IV intracellular and extracellular levels. Expression of V523del mutant, however, did partially restore both Col I / IV intracellular and extracellular levels to those of rWT condition." Indeed, on the basis of the quantification, it appears that for all the CIC-5 mutants the level of Col I/IV remains elevated compared CIC-5 rWT. It would be more convincing to also include in these westernblotting experiments, the condition shown in figure 1A and 1C (ctr and KD).

*We now include a representative western blot with all conditions for Collagen IV intracellular and extracellular levels (Control, *CLCN5* KD, CIC5-rWT and the three mutants) in new **supplementary figure 2A**. All data presented here come from the quantification of at least three independent experiments. We are now including the number of experiments (N) in all figure legends.*

4- Figure 1 E. Confocal analysis of ColI and IV expression must be quantified by specifying in the figure legend or in the material and methods sections the number of experiments and the number of cells analyzed for quantification.

We have included the number of experiments and cells analyzed in all figure legends.

5- Figure 1G. Please provide details of quantification (number of fields or cells quantified? How did you identified individual collagene fibers ?...).

We have added the details of quantification in material and methods section.

“To determine the number, volume and area of LAMP1, Col I and Col IV positive elements, we used the 3D objects counter v2.0 tool from FIJI (24). All images for each experiment were taken on the same day under the same conditions and the same z-step (0.5 μm), a minimum of 3 fields for each experiment ($N > 3$) were analysed. The threshold was automatically set for each image. DAPI was used to count the number of nuclei per field. Regarding quantification of collagen fibres, Col I fibres were defined as those particles with an area $> 2 \mu\text{m}^2$, perimeter $> 10 \mu\text{m}$; whereas Col IV fibres were defined as those Col IV particles with an area $> 1.5 \mu\text{m}^2$, perimeter $> 10 \mu\text{m}$.”

6- Figure 2A and 2B, could the author explain why they used KDEL as an ER marker, and not a more conventional marker such as calnexin? And wouldn't it be more rigorous to analysis the Manders's coefficient specifically for 40-50 individual cells per condition? The figure legend does not mention the number of fields or cells taken into account for the quantification shown in figures 2C and 2F.

The sequence Lys-Asp-Glu-Leu (KDEL) is known to be necessary for ER retention of proteins. Thus, antibodies against this sequence are commonly used to stain the ER (e.g., doi.org/10.15252/embr.202254701; doi.org/ 10.1096/fj.202100303R). We have now included a more detailed description of Manders' coefficient analysis by JACoP plugin (FIJI), stating that a minimum of three different fields (> 5 cells) for three independent experiments per condition were analysed.

“Manders' overlap coefficient was calculated using ImageJ plugin JACoP (24) in a minimum of three different fields (> 5 cells/field) for three independent experiments per condition. JACoP plugin measures colocalization using different indicators (Pearson's coefficient, Overlap coefficient or Manders' coefficients) on two images for all z-stacks.”

7- Figure 3C and 3D, once again, how many experiments were performed? This is not specified in the figure legend. It is difficult for the reviewer to be convinced based on the difference in tubulin expression. Especially in figure 3C for the line 3 for the blot ColIV, and on figure 3D, lanes 3 and 4 for the lysate samples. In addition, the signal for col I in figure 3C is very low in lane 1 and 2 (here, a membrane with a higher exposure should be shown.

We apologize for this. As stated above, all data presented here have been obtained from a minimum of 3 independent experiments. We have now included the number of experiments in each figure legend. We include new blots for Figure 3C and 3D.

8- On the basis of the data presented in figure 3, the reviewer believe that at this stage, there is an over-interpretation of the result when the authors state lines 287 and 288 that "These data strongly suggest that CIC-5 impairs collagens degradation by altering lysosomal acidification.

We thank the reviewer for this comment, we have removed “strongly” to avoid overinterpretation of our data.

9- To reinforce the results suggesting a defect in lysosome degradative activity upon CIC-5 KD, it is important to use other specific assay to test whether the degradation of other substrates is also compromise (ex Magic red assay, or cathepsin L or D activity assay...)

As suggested by the reviewer, we have used a Magic Red™ Cathepsin B Kit (#ICT938, BioRad) to test lysosomal degradative activity. Our data confirm a decrease of this activity on CIC-5 KD cells (30% reduction), which is in agreement that lysosomal function is impaired in these cells, leading to an overaccumulation of collagens. Besides, we have found that treatment with bafilomycin A1 reduced Magic Red fluorescence to a similar levels for all conditions, confirming its value as a normalization point for Lysosensor experiments. These data is included in new Figure 4 and the corresponding part in the Results section.

“In order to reinforce these results, we also tested lysosomal enzymatic activity using a Magic Red™ Cathepsin B Kit (#ICT938, BioRad). Our data revealed a 30% decrease in Magic Red fluorescence in CLCN5 KD compared to control cells, which was rescued in CIC-5 rWT cells. Treatment with Bafilomycin A1 (as positive control), reduced lysosomal activity to a similar level for all conditions (35% decrease of Magic Red fluorescence) (Figure 4B).”

10- Figure 4C, what is the rational for assessing the colocalization of LAMP1 with KDEL? In terms of lysosome distribution and intracellular location, lysosome are generally known to be more concentrated in the perinuclear area, and it has been reported that less acidic lysosomes are more peripheral. So, taking into account the effect of CIC-5 KD on lysosomal pH as shown in figure 4A, the quantification of perinuclear and peripheral lysosomal would be more relevant than the colocalization with KDEL.

By assessing the colocalization of LAMP1 with KDEL, we wanted to strengthen the idea that degradation of newly synthesized collagens (possibly misfolded) is impaired, thus even when they are targeted for lysosomal degradation, they accumulate in lysosomes that will be closer to the ER. It is important to state that collagens, due to their size, do not fit classic COP-II vesicles and need bigger carriers to be transported, a process that requires TANGO1 (Raote et al, 2018). Indeed, it has been shown that a subset of procollagen molecules are transported directly to the lysosomes from the ER for degradation to remove excess collagen from cells (Omari et al, 2018). As suggested by the reviewer, we have studied lysosome distribution in our cell lines. The results reveal a higher concentration of lysosomes in the perinuclear area in CIC-5 KD cells compared to control or rWT cells. Although these results do not fit other studies reporting that less acidic lysosomes are more peripheral (Johnson et al, 2016), we suggest that this increase in perinuclear lysosomes in cells lacking functional CIC-5 is the reflection of the accumulation of collagens in lysosomes (sitting near the ER), which cannot be degraded. We have added these new data in Figure 4, and included a new paragraph in the discussion.

RESULTS

“Next, we studied whether localization, size or number of lysosomes were altered in cells lacking CIC-5, which could be affected due to impaired lysosomal degradation of collagens. Immunofluorescence studies using anti-LAMP1 antibody and anti-KDEL antibody revealed a higher colocalization between both markers in CLCN5 KD cells than in control or rWT cells (2-fold increase, $p < 0.05$) (Figure 4C-D). Although the total number of lysosomes was not statistically different between cell lines, further analysis of their distribution revealed a higher concentration of lysosomes in the perinuclear area of CLCN5 KD cells compared to control and CIC-5 rWT cells (55% vs. 44% and 42%, respectively, $p < 0.01$) (Figure 4E). Notably, our results showed

increased percentage of big lysosomes in *CLCN5* KD cells compared to control or rWT cells (29.2±1.7%, 18.6±2.3% and 19.4±0.4%, respectively) (**Figure 4F**). Volume measurement also showed that lysosomes from *CLCN5* KD cells were bigger than control or rWT cells (Median volume of KD: 0.18 μm^3 , control: 0.11 μm^3 , rWT: 0.10 μm^3 , $p < 0.01$). These results fit with impaired degradation of collagens, which accumulate intracellular resulting in bigger lysosomes that remain closer to ER.”

DISCUSSION

“We deeply studied the role of CIC-5 on this increase in collagens production. Our findings revealed that this is the consequence of enhanced synthesis and impaired degradation. First, we have found that Col I/IV are mostly degraded lysosomally in RPTEC/TERT1 cells, a process that is altered in *CLCN5* KD cells. In fact, and contrarily to mainstream dogma (8), our results show that *CLCN5* depletion does affect the lysosomal pH. One possibility we cannot fully discard is that by affecting the endolysosomal system pH, deletion of *CLCN5* is not directly affecting collagens’ degradation, but their targeting to lysosomes. Importantly, collagen I and IV are big molecules with little flexibility that do not fit classic COP-II vesicles and need bigger carriers to be transported out of the ER, a process that requires TANGO1 (43). It has been shown that a subset of procollagen molecules are transported directly to the lysosomes from the ER for degradation to remove excess collagen from cells (44). Our results, indeed, also support this as a significant percentage of total collagens is located in lysosomes of *CLCN5* KD, which are bigger and closer to the ER. Moreover, we have found that inhibition of lysosomal acidification by Bafilomycin A1 increases localization of Col I and IV at lysosomes and decreases their colocalization with Golgi markers, supporting our hypothesis that defects in lysosomal pH led to collagens’ accumulation in these organelles. Further analysis of lysosomes’ distribution revealed a higher concentration at the perinuclear area in CIC-5 depleted cells. Although less acidic lysosomes have been reported to be more peripheral (45), in our case we suggest that this increase of lysosomes with higher pH at the perinuclear region of *CLCN5* KD cells is the reflection of impaired collagens’ degradation. Collagens that are transported from the ER to the nearby lysosomes for degradation cannot be removed and accumulate inside. Additionally, the dedifferentiation process previously shown by our group (18) due to CIC-5 lack of function is another possibility that could explain this change in lysosomal distribution.”

11- Line 299, authors indicate that the number of lysosomes was studied, but there are no results in figure 4 showing this analysis.

We apologize for this mistake. Figure 4E shows the relative number of lysosomes by sizes (divided by total). The total number of lysosomes was not statistically different between the different conditions. We have included this sentence in the results.

“Although the total number of lysosomes was not statistically different between cell lines, further analysis of their distribution revealed a higher concentration of lysosomes in the perinuclear area of *CLCN5* KD cells compared to control and CIC-5 rWT cells (55% vs. 44% and 42%, respectively, $p < 0.01$) (**Figure 4E**).”

12- Figure 5A, the effect of TGFbeta on col 1 and 4 mRNA expression is only tested in control cells. However, it is essential to test whether TGFbeta also increases col 1 and 4 mRNA expression in CIC-5 depleted cells in order to assess if TGFbeta and CIC-5 are involved in the same pathway (as it is done on figure 5D for protein expression).

We have studied Col I and IV mRNA expression in control, CIC-5 KD and rWT cells treated with vehicle or TGF- β . Our data, now included in Figure 5A, shows a clear synergic effect between TGF- β and CIC-5 depletion on Col I and IV mRNA expression. A possible explanation is that in CIC-5 cells, since MUC1 is not at the PM and β -catenin is released and translocated to the nucleus, TGF- β activates other pathways

(such as SMAD2/3) that potentiate collagens' transcription. In control cells, beta-catenin is at the PM, sequestered by MUC1 and TGF- β effect is milder.

13- The immunofluorescence results in figure 5B suggesting that upon TGF beta treatment or CIC-5 KD, beta-catenin is translocated from the plasma membrane to the nucleus should be reinforced by a biochemical approach using cytoplasmic and nuclear fractions from which beta-catenin expression is quantified by westernblotting.

As suggested by the reviewer, we have performed a fractionation assay to further confirm beta-catenin nuclear translocation in the presence or absence of TGF- β . These data fits with the results and quantification of IF images. This is included in the new Figure 5D.

RESULTS

“Immunofluorescence analyses revealed that, as expected, β -catenin is mostly localized at PM in control cells and is translocated to the nucleus after stimulation by the classic pathway activator TGF- β (35) (1.7-fold increase, $p < 0.01$). Interestingly, β -catenin was also more often found at the nucleus in *CLCN5* KD cells compared to control cells, which was prevented by CIC-5 rWT expression (KD: 1.5-fold increase, $p < 0.01$; rWT: 0.9-fold increase, n.s., compared to control cells). Treatment with TGF- β triggered higher nuclear translocation of β -catenin in CIC-5 rWT cells (1.7-fold increase, $p < 0.01$), but did not cause a major effect in CIC-5 depleted cells (**Figure 5B**, quantification in **5C**). In order to validate these data on β -catenin nuclear translocation, we performed a fractionation assay of control, *CLCN5* KD and CIC-5 rWT cell lysates. Our findings showed a clear increase of β -catenin in the nuclear fraction of *CLCN5* KD cells (Control: 24.7%, KD: 35.7%, rWT: 29.8%, $p < 0.05$) or TGF- β treated cells compared to control and CIC-5 rWT cells (Control treated: 43.2%, KD treated: 36.8%, rWT treated: 37.6%, $p < 0.05$) (**Figure 5D**).”

14- Figure 6B, 6E and 6F, the loading control (tubulin) is extremely variable between the different conditions tested. It would be advisable to use another loading control such as actin or GAPDH to ensure that the same quantity of protein extract is loaded.

We agree and apologize for this. We now include a better representative western blot.

Reviewer #2

The authors present an interesting manuscript describing the potential role of *Clc-5* in controlling cellular collagen homeostasis. The data suggest that *Clc-5* functions in acidifying the endolysosomal system and a loss of function results in two complementary changes that affect the levels of collagen types I and IV. First, the trafficking of cell surface mucin 1 is affected, which results in altered signalling. Beta catenin migrates more to the nucleus, leading to increased transcription of collagens. Second, the actual degradation of collagen is reduced, since the lysosomes not acidify appropriately. The data and model are interesting and would be of considerable interest to several researchers in diverse fields of cell biology and physiology. I have a few minor suggestions or concerns.

1. In the first figure, it is not clear if cycloheximide was used in the experiments to assay secretion. Please repeat secretion and degradation time courses in the presence of cycloheximide to remove the confounds from altered transcription and translation that the authors have demonstrated later. These data will go toward addressing another issue that might be exciting, which is whether the *clc-5* affects the actual processes of secretion - could it actually increase the secretory capacity of the cell to allow for the increased collagen?

Cycloheximide was used in the collagen time courses to assess the half-life of collagens. However, it was not used when assessing collagen secretion, so we agree that enhanced extracellular levels may be a reflection of higher production rather than increased secretory capacity. Indeed, data from our previous work (Duran et al, 2021), reveal that neither cTAGE5 or TANGO1 are altered in CIC-5 KD cells. In addition, we have studied other big secretory cargoes (such as Fibronectin 1), and our results, now included in new Figure 1, showed no differences in secretion. Altogether, these data suggest that CIC-5 does not modulate the secretory capacity of the cell.

2. It is unclear what the 'relative secreted levels of collagen' are in figure 1D. Is the secreted collagen normalised to account for the number of cells? A way to plot it could be to normalise the secreted collagen to the tubulin. Is this what has been done? Please clarify in the figure legend. Avoid plotting a ratio of secreted collagen to total collagen - this makes interpretation confusing, since both intracellular retention and secretion are differentially affected in these cells.

Yes, we have normalized secreted levels to tubulin levels. We have clarified this in the figure legend.

3. The differential effects of the loss-of-function mutants are interesting, but they raise some questions. E527D and I524K mutants express much more poorly than the WT protein. Could this reduced expression explain the observation that they did not rescue CLCN5 KD phenotype on Col I/IV intracellular and extracellular levels? Please repeat the experiment using far less CLCN5 WT DNA to induce equivalent low expression of wild type protein for a comparable experiment.

As discussed in Duran et al, 2021, all mutants are correctly expressed at the mRNA levels, although their protein levels vary compared to CIC-5 WT form likely due to enhanced degradation. To avoid any confusion, we have included qPCR data showing endogenous and exogenous levels of CIC-5, confirming that all mutants are correctly expressed (Supplementary Figure 1A). We also include one sentence in the text clarifying that mRNA levels are similar, although protein levels are more variable due to mutants being delivered for degradation (Duran et al, 2021). Nonetheless, as the reviewer suggests we cannot fully discard that the failure to rescue KD phenotype by CIC-5 mutants is due to the lower protein levels. We have included this in the discussion:

DISCUSSION

“Importantly, only the wild type form of CIC-5, but not the pathogenic variants, rescued *CLCN5* KD phenotype. In addition, and resembling phenotypic variability observed in DD1 patients (13), each CIC-5 mutant tested here had different rescue effects on Col I/IV release, as well as on MUC1 levels and trafficking. The differential recovery of *CLCN5* KD phenotype may correlate to the mutant expression levels, remaining CIC-5 function or intracellular localisation as shown before (18). For instance, it is not surprising that I524K mutant, which causes CIC-5 retention at the ER leading to their degradation (18), presented a phenotype more similar to *CLCN5* KD. On the other hand, V523del mutant that can traffic further intracellularly and presents complex glycosylation, is more similar to the wild type form. Interestingly, although the mutants cannot fully restore MUC1 levels to control situation, their effect on collagens is more variable. A possible explanation is that even low levels of MUC1 are sufficient to sequester β -catenin at the plasma membrane therefore preventing enhanced transcription of Col I/IV. Altogether, our results are a first step to understand renal fibrosis development in DD1 patients, and future studies on different CIC-5 mutants are needed to better assess their contribution.”

4. Previous results from the same authors have shown that E527D and V523del have similar intracellular distributions, while only the I524K was retained in the ER. This is not consistent with the current explanation for the different phenotype with V523del. Please clarify.

In our previous study (Durán et al, 2021), we showed that E527D mutant does reach the endosomes, while V523del mutant has lower colocalization with endosomal markers. These differences in intracellular distribution may explain the distinct phenotype regarding collagen secretion. Notably, an on-going project of our group is to understand how these mutations in CIC-5 can lead to major changes regarding localization and protein stability.

5. One way to clarify the suggestion that ER-retention of the Clc-5 prevents it from reaching its appropriate site of function, is to use a Clc-5 construct under the control of the RUSH system. Under baseline conditions, the protein should be retained in the ER. Once released (by addition of biotin), the protein should be transported to its correct location. Rescue of the phenotype can then be assayed with or without biotin.

We thank the reviewer for this suggestion. The main point of this manuscript is to understand how CIC-5 loss-of-function may promote progression to renal fibrosis, while studying mutants localization is out of the scope of this manuscript. As stated above, this is an interesting topic that our group is currently investigating to assess how mutations closely located can lead to big differences in terms of CIC-5 intracellular trafficking.

6. The 'number of fibres' is a confusing depiction of the extracellular collagen. Could the authors please present the total fluorescence associated with the extracellular collagen as well?

We have added the details of quantification of collagen fibers in material and methods section. In addition, we now include the total fluorescence associated with extracellular collagen as suggested by the reviewer. Our new data agree with WB results, showing a higher than 2.5 fold increase in Col I / IV extracellular fluorescence in CLCN5 KD cells compared to control or CIC-5 rWT cells. We have included this data in new figure 1G and referenced it accordingly in the text.

RESULTS

“In accordance with WB results, we found that *CLCN5* KD cells present a higher than 2.5 fold increase in Col I / IV extracellular fluorescence compared to control and rWT cells (**Figure 1F**, quantification in **1G**), which correlated with detection of more Col I / IV extracellular fibres (Col I: 11.7 vs. 2.7 and 3.3 fibres; Col IV: 13.5 vs. 5.2 and 2.0 fibres, KD vs. control and rWT cells respectively) (**Figure 1H**).”

7. The localization of collagen type I or IV to lysosomes is not clear in the images. Arrows supposedly indicating colocalization often actually point at sites where the collagen and LAMP1 signal are next to each other, but not colocalized. The difference is several hundred nanometres. Please provide better images, remove arrows toward sites of collagen / LAMP1 signal touching each other without colocalization, and highlight some key locations with a magnified inset.

As suggested by the reviewer, we have included better images with higher magnification that show the close proximity between Col I/IV and LAMP1. These images are included in new Figure 2 (For Col I) and new Supplementary Figure 3 (For Col IV).

8. In figure 3D, bafilomycin seems to have enhanced collagen IV degradation. What does this mean?

We agree that our results show a tendency of bafilomycin to reduce collagen IV intracellular levels in CIC-5 KD cells, although it is not altering significantly Col IV secreted levels. Since Bafilomycin inhibits vATPase function (and our results confirmed that impair lysosomal function), although we cannot completely discard that it is enhancing collagen degradation, we suggest that it is altering the trafficking of other components required for the synthesis of collagen. Another possibility is that Bafilomycin A1 treatment,

under CLCN5 KD background, promotes collagen degradation through other pathways like autophagy or proteasome. This is an exciting question; however, it is out of the scope of this work.

9. Inhibiting organelle acidification with bafilomycin has also been shown to affect secretory function and trap secretory cargoes at the Golgi apparatus or the endolysosomal network.

We thank the reviewer for this comment. We have now included immunofluorescence data in control cells treated with or without bafilomycin to show collagen localization at ER, Golgi and Lysosomes. First, our data showed minimum colocalization of collagens with Golgi marker GM130. Treatment with bafilomycin did not affect collagens colocalization with ER marker KDEL, but increased collagens colocalization with lysosomal marker LAMP1. Accordingly with collagens being stacked at lysosomes, colocalization with GM130 was lower in cells treated with bafilomycin. These data are included in new Figure 2, new Supplementary Figure 3 and new Supplementary Figure 4 and in the corresponding part in the text.

RESULTS

“Differentiated control, *CLCN5* KD and CIC-5 rWT cells were processed for immunofluorescence, co-stained with anti-Col I / IV and the endoplasmic reticulum (ER) marker KDEL, Golgi Apparatus marker GM130 or the lysosomal marker LAMP1, and imaged by confocal microscopy. Our results showed major colocalization of Col I and IV with KDEL in *CLCN5* KD cells compared to control or rWT cells (17.5- and 10.5- fold increase, respectively, $p < 0.001$) (Figure 2A, Supplementary Figure 3A). In addition, we also detected higher colocalization of Col I and IV with LAMP1 in *CLCN5* KD cells than control or CIC-5 rWT cells (3- and 3.5- fold increase, $p < 0.05$) (Figure 2B, Supplementary Figure 3B). Importantly, colocalization of both Col I with GM130 was similar in all conditions, although it was mildly higher in *CLCN5* KD cells for Col IV (Supplementary Figure 4A-B). Altogether, our results confirmed that collagens mostly accumulate in the ER of *CLCN5* KD, and at lower degree at the lysosomes, suggesting an effect on collagens’ degradation. To further assess this possibility, we also analysed collagens’ localisation after treatment with Bafilomycin A1 (inhibitor of v-ATPase that impairs lysosomal activity). Our results showed no major effects on Col I / IV colocalization with KDEL, but increased the colocalization with LAMP1 in control and CIC-5 rWT cells, and to a lesser extent in *CLCN5* KD cells (Figure 2A-B and Supplementary Figure 3A-B). Notably, Col I colocalization with GM130 was strongly reduced after Bafilomycin treatment, which fits with higher accumulation in lysosomes and impaired intracellular trafficking (Supplementary Figure 4A).”

10. Could the authors please show using immunofluorescence (as in figure 2D and 2E) that treatment with inhibitors of lysosomal activity actually results in an increase in colocalization of collagen and LAMP1? If the collagen is retained in the secretory pathway, this might explain why bafilomycin has minimal effect on degradation in the lysosome.

As suggested by the reviewer, we have studied Collagen colocalization with LAMP1 in bafilomycin treated cells. Our data showed increased collagens colocalization with lysosomal marker LAMP1 in cells treated with bafilomycin. These results are included in new figure 2 and supplementary figure 3. We have extended this part in the results and discussion sections.

RESULTS

“Differentiated control, *CLCN5* KD and CIC-5 rWT cells were processed for immunofluorescence, co-stained with anti-Col I / IV and the endoplasmic reticulum (ER) marker KDEL, Golgi Apparatus marker GM130 or the lysosomal marker LAMP1, and imaged by confocal microscopy. Our results showed major colocalization of Col I and IV with KDEL in *CLCN5* KD cells compared to control or rWT cells (17.5- and 10.5- fold increase, respectively, $p < 0.001$) (Figure 2A, Supplementary Figure 3A). In addition, we also detected higher colocalization of Col I and IV with LAMP1 in *CLCN5* KD cells than control or CIC-5 rWT cells (3- and 3.5- fold increase, $p < 0.05$) (Figure 2B, Supplementary Figure 3B). Importantly, colocalization

of both Col I with GM130 was similar in all conditions, although it was mildly higher in *CLCN5* KD cells for Col IV (**Supplementary Figure 4A-B**). Altogether, our results confirmed that collagens mostly accumulate in the ER of *CLCN5* KD, and at lower degree at the lysosomes, suggesting an effect on collagens' degradation. To further assess this possibility, we also analysed collagens' localisation after treatment with Bafilomycin A1 (inhibitor of v-ATPase that impairs lysosomal activity). Our results showed no major effects on Col I / IV colocalization with KDEL, but increased the colocalization with LAMP1 in control and CIC-5 rWT cells, and to a lesser extent in *CLCN5* KD cells (**Figure 2A-B** and **Supplementary Figure 3A-B**). Notably, Col I colocalization with GM130 was strongly reduced after Bafilomycin treatment, which fits with higher accumulation in lysosomes and impaired intracellular trafficking (**Supplementary Figure 4A**)."

DISCUSSION

"We deeply studied the role of CIC-5 on this increase in collagens production. Our findings revealed that this is the consequence of enhanced synthesis and impaired degradation. First, we have found that Col I/IV are mostly degraded lysosomally in RPTEC/TERT1 cells, a process that is altered in *CLCN5* KD cells. In fact, and contrarily to mainstream dogma (8), our results show that *CLCN5* depletion does affect the lysosomal pH. One possibility we cannot fully discard is that by affecting the endolysosomal system pH, deletion of *CLCN5* is not directly affecting collagens' degradation, but their targeting to lysosomes. Importantly, collagen I and IV are big molecules with little flexibility that do not fit classic COP-II vesicles and need bigger carriers to be transported out of the ER, a process that requires TANGO1 (43). It has been shown that a subset of procollagen molecules are transported directly to the lysosomes from the ER for degradation to remove excess collagen from cells (44). Our results, indeed, also support this as a significant percentage of total collagens is located in lysosomes of *CLCN5* KD, which are bigger and closer to the ER. Moreover, we have found that inhibition of lysosomal acidification by Bafilomycin A1 increases localization of Col I and IV at lysosomes and decreases their colocalization with Golgi markers, supporting our hypothesis that defects in lysosomal pH led to collagens' accumulation in these organelles."

11. In section 5, TGF- β treatment increases collagen mRNA in control cells. Please repeat this in the other two cell lines (both the depletion as well as the rescue). This will be useful to look at in light of the rest of the data in the same figure.

We have studied Col I and IV mRNA expression in control, CIC-5 KD and rWT cells treated with vehicle or TGF- β . Our data, now included in Figure 5A, shows a clear summatory effect between TGF- β and CIC-5 depletion on Col I and IV mRNA expression. A possible explanation is that in CIC-5 cells, since MUC1 is not at the PM and β -catenin is released and translocated to the nucleus, TGF- β activates other pathways (such as SMAD2/3) that potentiate collagens' transcription. In control cells, beta-catenin is at the PM, sequestered by MUC1 and TGF- β effect is milder.

12. The text describing supplementary figure 3 needs to be corrected - the data are in supplementary figure 4.

We thank the reviewer for pointing this out. This is now corrected.

13. The data from transgenic mice are a nice addition. Could the authors also carry out a similar staining for mucous? PAS staining should reveal any alterations in the mucous thickness. Similarly, an immunohistochemical staining of beta catenin to compare surface and nuclear localisation as in the cell lines.

We thank the reviewer for this suggestion. We have performed MUC1 staining in mice slices, however MUC1 is mostly expressed in distal tubule epithelial cells or in injured tissue in mice. Our analyses did not show major differences regarding MUC1 levels. Similarly, PAS staining, which nicely stains proximal tubules

brush border did not reveal any major alteration. We consider that the lack of major effects in the brush border or MUC1 levels is due to mice being young, thus they did not accumulate enough damage to present these alterations. Unfortunately, Dr. Baisong Lu has discontinued his colony of Clcn5 KO mice and we do not have access to renal tissue of aged mice. Although having these data would add more information, our results on collagen levels and the membrane thickness from mice tissue, which are clearly pointing towards a pre-fibrotic phenotype, strongly support the role of CIC-5 on renal fibrosis.

14. As a final small issue - the text is confusing on one point. The authors suggest "our data indicates that lack of CLCN5 promotes collagen synthesis through β -catenin pathway and, in combination with lysosomal degradation impairment, provokes a huge intracellular collagen accumulation that will also end in increased release of collagens to the extracellular medium." That statement (made several times) suggests that increased secretion could come from either increased synthesis or the collagen that is stuck in lysosomes. This would be very interesting as well, if it were demonstrated. From the evidence presented, it would seem clear that the collagen retained in poorly-acidified lysosomes does not get secreted (for instance supp fig 3A, Bafilomycin). The increase in secretion must come entirely from the increased synthesis. There are no other experiments testing the hypothesis that collagen retained in the lysosomes is secreted. Please rewrite to clarify this - from the abstract to the discussion. Alternatively, authors could carry out experiments to test whether the lysosomal accumulations of collagen are secreted, though that is not in the ambit of this paper.

We have rewritten extensively the text as suggested by the reviewer.

Reviewer #3

The authors present interesting data regarding the effects of CIC-5 knocked down on cellular synthesis and secretion of collagens. The implication is that this mediated renal fibrosis and eventual chronic kidney disease/kidney failure in patients with Dent disease type 1. The fibrosis aspect of the observation is reproduced in a knock out mouse model. Overall, the data are very compelling in the cell culture experiments. Multiple experiments are presented that demonstrate abnormal lysosomal function in the CIC-5 knock down lines. Good evidence is presented to implicate the beta catenin pathway. The cellular phenotype can be rescued with wild type CIC-5 protein but not protein that harbors human pathogenic mutations.

1. For the most part, the paper relies on observations in a single line of culture cells. It would be ideal to see this reproduced at least in certain aspects in an independent model system.

We have used renal proximal tubule epithelial cells (RPTEC), which are the gold standard to study the functioning of the proximal tubule epithelia. These cells maintain most of the characteristics of the proximal tubule, such as primary cilia, high endocytic capacity, CIC-5 expression and cell polarity (e.g., MUC1 or Megalin/Cubilin only at the apical membrane), as shown before (Wieser et al, 2008). It is important to state that we do not rely on one single cell line, but we have generated several lines with RPTEC depleted of CIC-5 and expressing different versions of the protein (WT or mutants) (Durán et al, 2021). In addition, our data from Dent Disease type I mice model clearly supports our results in cell lines.

2. A lot more could be done with the animal model to demonstrate key features in the proximal tubule consistent with the salt culture model. This might do some to alleviate concerns that there is any lack of generalizability of the cell culture studies.

We thank the reviewer for this suggestion. We have performed MUC1 staining in mice slices, however MUC1

is mostly expressed in distal tubule epithelial cells or in injured tissue in mice. Our analyses did not show major differences regarding MUC1 levels. Similarly, PAS staining, which nicely stains proximal tubules brush border did not reveal any major alteration. We consider that the lack of major effects in the brush border or MUC1 levels is due to mice being young, thus they did not accumulate enough damage to present these alterations. Unfortunately, Dr. Baisong Lu has discontinued his colony of Clcn5 KO mice and we do not have access to renal tissue of aged mice. Although having these data would add more information, our results on collagen levels and the membrane thickness from mice tissue, which are clearly pointing towards a pre-fibrotic phenotype, strongly support the role of CIC-5 on renal fibrosis.

3. Did the authors look at any effects within the podocytes of their culture cells in order to see if similar events could be going on there to explain the focal global sclerosis that is been described in Dent disease.

Our cell lines are derived from the proximal tubule, and they do not have any features related to the podocytes. As suggested by the reviewer, we have analysed the glomeruli in mice slices, which revealed no major differences between the WT and KO.

4. At a minimum, it seems the paper could benefit from a good strengths and weaknesses section in the discussion to point out some of the things that might need to be verified in either additional cell culture systems, organoids, or intact animals.

We thank the reviewer for this comment. Accordingly, we have included a new paragraph in the discussion regarding future research in organoids or other DD animal models.

DISCUSSION

“Future research involving aged mice would strengthen these results as renal fibrosis in DD1 patients is a progressive process and is not normally observed during childhood (25). Besides, further investigation using patient derived organoids could provide more information regarding the involvement of CIC-5 in renal fibrosis than cell culture systems. Another option would be to develop co-culture systems involving renal proximal tubule cells, endothelial cells and stromal cells to provide the proper microenvironment and ECM production (48).”

We hope these comments are satisfactory and the manuscript is suitable for publication in Life Science Alliance Journal.

Sincerely yours,

Gerard Cantero Recasens
Principal Investigator
Renal Physiopathology Group
Vall d’Hebron Research Institute (VHIR)
Pg. Vall d’Hebron 119-129
08035 Barcelona (Spain)

April 9, 2024

RE: Life Science Alliance Manuscript #LSA-2023-02444-TR

Dr. Gerard Cantero-Recasens
Vall d'Hebron Institut de Recerca
Renal Physiopathology Group
Pg. Vall d'Hebron 119-129
Barcelona 08035
Spain

Dear Dr. Cantero-Recasens,

Thank you for submitting your revised manuscript entitled "Renal antiporter CIC-5 regulates collagen I/IV through β -catenin pathway and lysosomal degradation". We would be happy to publish your paper in Life Science Alliance pending final revisions necessary to meet our formatting guidelines.

- please be sure that the authorship listing and order is correct
- please upload all figure files as individual ones, including the supplementary figure files; all figure legends should only appear in the main manuscript file
- please note that the titles in the system and manuscript files must match
- the full first and last name (middle names as initials) of each author should be given on the title page
- please consult our manuscript preparation guidelines (<https://www.life-science-alliance.org/manuscript-prep>) and ensure your manuscript sections are in the correct order and labeled correctly-the background should be an introduction, etc.
- please incorporate any points from the Conclusions section into the Discussion; we only allow a Discussion section
- please be sure that all co-authors are listed in the Author Contributions section in the manuscript text
- please move your main, supplementary figure, and table legends in the main manuscript text after the references section
- please add callouts for Figures 5H and 8A-B to your main manuscript text;

FIGURE CHECKS:

- please add sizes next to the blots in Figure 3

A. FINAL FILES:

B. MANUSCRIPT ORGANIZATION AND FORMATTING:

Sincerely,

Reviewer #1 (Comments to the Authors (Required)):

The authors responded convincingly to my concerns and provided robust new data in support of their conclusion.

Reviewer #2 (Comments to the Authors (Required)):

The authors have addressed my concerns. I have no further suggestions.

Reviewer #3 (Comments to the Authors (Required)):

My comments have been addressed. Thanks!

April 17, 2024

RE: Life Science Alliance Manuscript #LSA-2023-02444-TRR

Dr. Gerard Cantero-Recasens
Vall d'Hebron Institut de Recerca
Renal Physiopathology Group
Pg. Vall d'Hebron 119-129
Barcelona 08035
Spain

Dear Dr. Cantero-Recasens,

Thank you for submitting your Research Article entitled "Renal antiporter CIC-5 regulates collagen I/IV through β -catenin pathway and lysosomal degradation". It is a pleasure to let you know that your manuscript is now accepted for publication in Life Science Alliance. Congratulations on this interesting work.

DISTRIBUTION OF MATERIALS:

Again, congratulations on a very nice paper. I hope you found the review process to be constructive and are pleased with how the manuscript was handled editorially. We look forward to future exciting submissions from your lab.

Sincerely,
